# Amortizing Maximum Inner Product Search with Learned Support Functions

**Theo X. Olausson** [1]  **João Monteiro** [1]  **Michal Klein** [1]  **Marco Cuturi** [1]

## Abstract

Maximum inner product search (MIPS) is a crucial subroutine in machine learning, requiring the identification of a vector taken within a database (the keys) that best aligns with a given query. We propose *amortized MIPS*: a regression-based approach that trains neural networks to directly predict MIPS solutions, amortizing the cost of repeatedly solving MIPS for queries drawn from a known distribution over a fixed key database. Our key insight is that the MIPS value function is the *support* function of the set of keys, a well-studied convex function whose gradient yields the optimal key. This motivates two complementary amortized models: SUPPORTNET, an input-convex neural network trained to regress the support function, and KEYNET, a vector-valued network that directly regresses the optimal key. SUPPORTNET can serve as a *cluster router*, steering queries toward relevant database partitions, while KEYNET can be used as a *drop-in replacement* for the original query, fed directly to off-the-shelf indexing pipelines. Our experiments on the BEIR benchmark show that, for document embeddings, learned SUPPORTNETs and KEYNETs significantly improve IVF match rates when accounting for compute effort, whether measured in FLOPs, number of probes, or wall-clock time. Our code is available at: https://github.com/apple/ml-amips.

## 1. Introduction

Maximum inner product search (MIPS) is the workhorse of retrieval. For a query vector $\mathbf{x} \in \mathbb{R}^d$ and a database of vectors $\mathcal{Y} = \{\mathbf{y}_1, \ldots, \mathbf{y}_n\} \subset \mathbb{R}^d$, MIPS identifies the

vector in $\mathcal{Y}$ that with highest dot-product with $\mathbf{x}$:

$$\mathbf{y}^\star(\mathbf{x}) = \arg\max_{\mathbf{y} \in \mathcal{Y}} \langle \mathbf{x}, \mathbf{y} \rangle. \qquad (1)$$

While this problem is typically solved exhaustively by leveraging GPU parallelism in $O(nd)$ time, this becomes computationally prohibitive when dealing with large-scale datasets containing millions of high-dimensional vectors. Yet, MIPS plays a crucial role in various domains, including recommendation systems (Bachrach et al., 2014; Koenigstein et al., 2012), information retrieval (Shrivastava & Li, 2014), retrieval-augmented generation (Lewis et al., 2020), and large-scale classification (Dean et al., 2013; Yu et al., 2014).

**Approximate MIPS.** The computational challenge posed by large-scale MIPS has motivated extensive research into *approximate* MIPS methods, which trade a small amount of accuracy for significant computational savings. These approaches typically rely on indexing structures (hashing, trees, graphs) or quantization schemes that must be queried at inference time (see Section 2). While effective, these approaches make no (or limited) use of the query distribution (e.g. by optimizing quantization lookups or criteria involving linear transforms).

**Amortized MIPS: A Learning Perspective.** We propose a fundamentally different approach: Rather than building query-agnostic indexing structures, we train networks to directly predict MIPS solutions, amortizing the computational cost of search across queries drawn from a known distribution $p_{\mathcal{X}}$. The key insight underlying our approach is that the MIPS *value function*, the max inner product, is the *support function* of the database $\mathcal{Y}$:

$$\sigma_{\mathcal{Y}}(\mathbf{x}) = \max_{\mathbf{y} \in \mathcal{Y}} \langle \mathbf{x}, \mathbf{y} \rangle. \qquad (2)$$

This function is convex (as the pointwise maximum of linear functions) and positively 1-homogeneous. Moreover, by the envelope theorem, its gradient at each query equals the optimal database vector: $\nabla \sigma_{\mathcal{Y}}(\mathbf{x}) = \mathbf{y}^\star(\mathbf{x})$. These structural properties of the support function motivate two learning paradigms: either train a network to predict the real-valued function (SUPPORTNET), or to directly predict the top key (KEYNET). Figure 1 situates these two views against the

---

*TXO is now at MIT; work done during an internship at Apple. [1]Apple. Correspondence to: Theo X. Olausson <theoxo@csail.mit.edu>, João Monteiro <jmonteiro2@apple.com>, Michal Klein <michalk@apple.com>, Marco Cuturi <cuturi@apple.com>.

*Proceedings of the 43$^{rd}$ International Conference on Machine Learning*, Seoul, South Korea. PMLR 306, 2026. Copyright 2026 by the author(s).

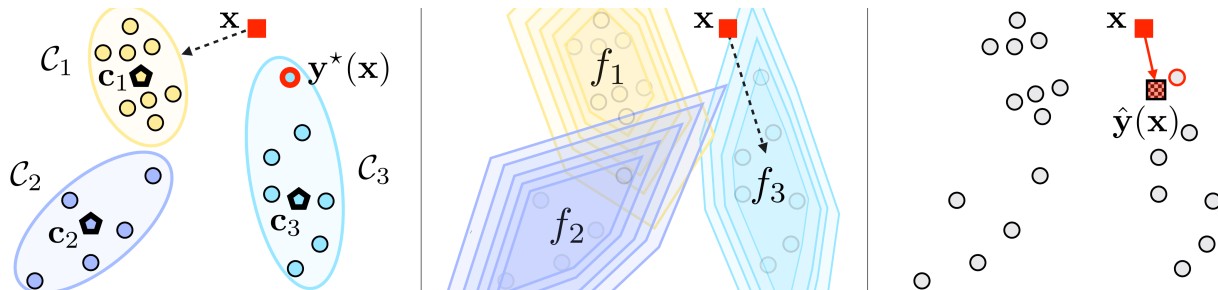

**Figure 1. Two complementary uses of amortized MIPS.** *(left)* the default coarse step in inverted-file indices scores the query $\mathbf{x}$ against each cluster centroid $\mathbf{c}_i$ to decide the highest-scoring cell. Here the nearest centroid would be routed to the top-left cluster, but the true top key is an outlier near the top of the vertically-stretched rightmost cluster, so centroid routing picks the wrong cell. *(middle)* SUPPORTNET learns one support function $f_i$ per cluster. Each support function is nonlinear, and quantifies, for any point $\mathbf{x}$, the value of the highest dot-product between $\mathbf{x}$ and any of the points within the cluster, correctly selecting the cluster with the best attainable match even when its centroid is less aligned with $\mathbf{x}$: $f_3(\mathbf{x}) > f_1(\mathbf{x})$ despite $\mathbf{x}^T\mathbf{c}_1 > \mathbf{x}^T\mathbf{c}_3$. *(right)* KEYNET is trained to predict a vector $\hat{\mathbf{y}}(\mathbf{x})$ that is as close as possible to the true top-1 key $\mathbf{y}^\star(\mathbf{x})$, and can be used as a drop-in replacement for $\mathbf{x}$ to any off-the-shelf index approach.

centroid-routing coarse step at the heart of inverted-file indices, which can route a query to a cell that does not contain its true top key whenever the centroid baseline is a poor predictor of within-cluster maximum similarity. SUP-PORTNET's scalar outputs make it well-suited as a *cluster router* that steers queries toward relevant database partitions, while KEYNET's vector output can be used as a *drop-in replacement* for the original query that can be fed directly to off-the-shelf indexing pipeline. We propose to ground the training of SUPPORTNET on loosely-constrained input convex neural networks (ICNNs; Amos et al. 2017), to learn the convex support function as $f_\theta(\mathbf{x}) \approx \sigma_{\mathcal{Y}}(\mathbf{x})$. In principle, an approximation of the optimal key could be recovered via a backward gradient computation (w.r.t. $\mathbf{x}$) as $\nabla_\mathbf{x} f_\theta(\mathbf{x})$. Training KEYNET can be done instead by directly regressing the optimal key $F_\theta(\mathbf{x}) \approx \mathbf{y}^\star(\mathbf{x})$, bypassing backward computation entirely at inference. It is important to note that our methods leverage *amortized optimization* (Amos, 2023b): they assume access to a relevant distribution of queries $p_\mathcal{X}$ from which vectors $\mathbf{x}$ can be sampled from, along with their ground truth score $\sigma_{\mathcal{Y}}(\mathbf{x})$ and ground-truth key $\mathbf{y}^\star(\mathbf{x})$, recovered by running MIPS beforehand to define a *dataset* used to train our neural networks on. Our contributions are:

- We introduce SUPPORTNET and KEYNET, two models and their regression-based training approach to amortize AMIPS. We design loss functions for both: score regression and gradient matching for the first, multivariate regression and *score consistency* –based on Euler's theorem for homogeneous functions– for the latter.
- We propose *multi-task* extensions of these two approaches when learning multiple MIPS tasks *jointly*, using parameter sharing. This is useful to address problems where the keys can be partitioned into clusters, but are queried using queries sampled from the same distribution. We show how these support functions learned jointly can be used to iden-

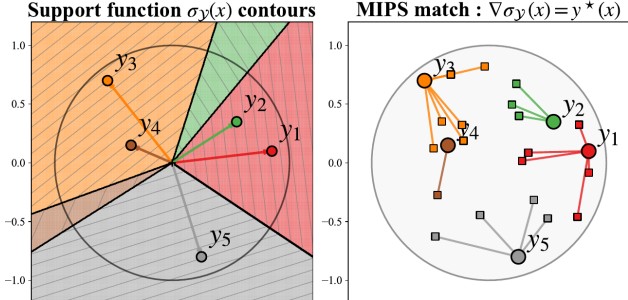

**Figure 2.** In this figure, the set $\mathcal{Y}$ of keys (shown as dots) consists of 5 points. The support function, represented as a contourplot on the left, is a convex, piecewise-linear function of the query $\mathbf{x}$. The gradient $\nabla\sigma_{\mathcal{Y}}(\mathbf{x})$ at any query equals the database point $\mathbf{y}^\star$ that maximizes $\langle\mathbf{x}, \mathbf{y}\rangle$. The gradient $\nabla_\mathbf{x}\sigma_{\mathcal{Y}}(\mathbf{x})$ is exactly the key to which a query is matched (here represented as small squares). While SUPPORTNET is trained to model the contour plot on the left (modeling a piecewise affine function), KEYNET is trained to map inputs to optimal keys directly, as shown on the right plot (a piecewise constant vector-valued function).

tify to which cluster a query should be matched, without running any comparisons with the keys in those clusters.

- We demonstrate on retrieval benchmarks that amortized MIPS achieves high match rates, and that modifying queries using learned models improves recall of standard approximate search indices. We consider two scenarios where support function and top-key predictors are respectively highlighted: We cluster the keys into subgroups and train (jointly) as many support functions, and use this as a routing mechanism; we convert a query into its predicted key and test whether fast AMIPS search with the predicted key returns better results than when run with the original query. In both cases we observe significant speedups (factoring accuracy vs. FLOPs) across a wide range of hyperparameters for our architectures.

## 2. Background

**Maximum Inner Product Search.** The exact search to compute either (2) or (1) requires $O(nd)$ time, which can be prohibitive for large-scale applications. Approximate MIPS methods fall into several categories. *Transformation-based* approaches convert MIPS to nearest neighbor search via augmented embeddings (Shrivastava & Li, 2014; Bachrach et al., 2014; Neyshabur & Srebro, 2015). The key insight is that inner products can be encoded as Euclidean distances in a higher-dimensional space, enabling the use of efficient nearest neighbor algorithms. *Quantization methods* compress database vectors for fast approximate inner products (Jégou et al., 2011; Guo et al., 2016; 2020). Product quantization and its variants decompose vectors into subvectors, each quantized independently, achieving significant compression while maintaining reasonable accuracy. *Graph-based* methods construct proximity graphs for greedy traversal (Morozov & Babenko, 2018; Malkov & Yashunin, 2020). These approaches build navigable small-world graphs where edges connect vectors with high mutual inner products, enabling efficient beam search at query time. *Learning-to-hash* approaches train neural networks to produce binary codes preserving inner product structure (Dai et al., 2020; Shen et al., 2015). Recent work has also explored learned indices that adapt to data distributions (Kraska et al., 2018; Vecchiato et al., 2024). These methods are largely query-agnostic at inference time, though some may partially leverage query statistics during index construction (Chen et al., 2024; Wei et al., 2025); our approach instead trains neural networks explicitly on the query distribution, encoding MIPS predictions directly into model weights.

**Input Convex Neural Networks (ICNNs).** ICNNs are neural architectures guaranteed to be convex in their inputs (Amos et al., 2017). An ICNN with $L$ layers computes:

$$\mathbf{z}_1 = \sigma\left(\mathbf{W}_0^{(x)}\mathbf{x} + \mathbf{b}_0\right) \in \mathbb{R}^h,$$

$$\mathbf{z}_{i+1} = \sigma\left(\mathbf{W}_i^{(z)}\mathbf{z}_i + \mathbf{W}_i^{(x)}\mathbf{x} + \mathbf{b}_i\right) \in \mathbb{R}^h, \ 1 \le i \le L-1,$$

$$f(\mathbf{x}) = \mathbf{z}_L \in \mathbb{R},$$

where $\sigma$ is a convex, monotonically non-decreasing activation (e.g., ReLU, softplus), and crucially, $\mathbf{W}_i^{(z)} \ge 0$ element-wise. The non-negativity ensures that each layer preserves convexity. The "passthrough" connections $\mathbf{W}_i^{(x)}\mathbf{x}$ (with unconstrained weights) provide expressiveness, allowing ICNNs to universally approximate continuous convex functions (Chen et al., 2019). To train ICNNs, the nonnegativity constraint on the $\mathbf{W}_i^{(z)}$ matrices can be either enforced using clipping (or a softer rectifier, e.g. softplus) so that it only contains non-negative values after each update, or loosely satisfied by adding a regularizer that penalizes negative values, such as $\sum_i \|\operatorname{ReLU}(-\mathbf{W}_i^{(z)})\|^2$. We use the

latter approach and combine it with the principled initialization proposed by Hoedt & Klambauer (2023), which initializes these weight matrices with non-negative values.

**Connection to Optimal Transport.** Brenier's theorem establishes that the optimal transport (OT) map between two probability distributions (under squared Euclidean cost) is always achieved by the gradient of a convex function. This has motivated substantial work on learning convex potentials to estimate optimal transport maps from samples (Makkuva et al., 2020; Korotin et al., 2021; Amos, 2023a; Bunne et al., 2022; Korotin et al., 2023). Conversely, that theorem also states that the gradient of a convex function is the optimal map linking any source distribution to the image of that distribution under the gradient. Our amortized MIPS problem therefore has an interesting connection to OT: Indeed, the mapping $\mathbf{x} \mapsto \mathbf{y}^\star(\mathbf{x}) = \nabla\sigma_{\mathcal{Y}}(\mathbf{x})$ is, by definition, the gradient of a convex function (the support function), making it a Brenier map. However, very much unlike the typical OT setting where one must *discover* that potential function, in MIPS we have direct access to ground-truth examples of the Brenier potential and its gradient: for any query $\mathbf{x}$, we can compute its optimal match $\mathbf{y}^\star(\mathbf{x})$ via exhaustive search, giving us a gradient, and the dot-product with $\mathbf{x}$, the potential. As a result, amortized MIPS can be seen as a supervised OT learning problem.

## 3. Methods: Neural Amortized MIPS

We propose *amortized maximum inner product search*: a learning-based approach that trains neural networks to directly predict MIPS solutions. Rather than building indexing structures or hashing schemes that must be queried at inference time, we train a neural network to encode information about the MIPS solution directly in its functional form, amortizing the computational cost of search across queries.

**Properties of Support Functions.** The MIPS value function (2), which gives the maximum inner product within a set of keys $\mathcal{Y}$ as a function of the query $\mathbf{x}$, is the *support function* of $\mathcal{Y}$. Support functions are fundamental, well-studied objects in convex analysis (Rockafellar, 1970; Hiriart-Urruty & Lemaréchal, 2004). For any set $\mathcal{Y}$, $\sigma_{\mathcal{Y}}$ is convex (as the pointwise maximum of linear functions) and positively 1-homogeneous: $\sigma_{\mathcal{Y}}(\alpha\mathbf{x}) = \alpha\,\sigma_{\mathcal{Y}}(\mathbf{x})$ for $\alpha > 0$. By the envelope theorem, the gradient of the support function at $\mathbf{x}$ equals the maximizing element: $\nabla\sigma_{\mathcal{Y}}(\mathbf{x}) = \mathbf{y}^\star(\mathbf{x})$. Figure 2 illustrates this relationship.

**Learning Targets.** This connection between the support function and its gradient motivates two complementary approaches: SUPPORTNET learns a (real-valued) convex potential that mimics the support function directly on samples that matter: $f_\theta : \mathbb{R}^d \to \mathbb{R}$ is such that $f_\theta(\mathbf{x}) \approx \sigma_{\mathcal{Y}}(\mathbf{x})$ for

**x** that *matter*, i.e. sampled according to $p_{\mathcal{X}}$. The optimal key can be then recovered via $\nabla_{\mathbf{x}} f_\theta(\mathbf{x})$ computed using automatic differentiation. KEYNET learns to produce instead, directly, a vector-valued function $F_\theta : \mathbb{R}^d \to \mathbb{R}^d$ that approximates the predicted optimal key $F_\theta(\mathbf{x}) \approx \mathbf{y}^\star(\mathbf{x})$ for **x** sampled according to $p_{\mathcal{X}}$. The support function can be then approximated as $\langle F_\theta(\mathbf{x}), \mathbf{x} \rangle$.

**Clustered Variants.** In many applications, a very large database $\mathcal{Y}$ can be naturally partitioned into $c$ clusters $\mathcal{Y}_1, \ldots, \mathcal{Y}_c$ (e.g. using $k$-means clustering). Rather than learning a single support function for each of these clusters, we propose to cast this joint learning as a multi-task learning problem in which a collection of support functions is learned by sharing parameters. As a result, the goal is to simultaneously learn the support functions for all clusters:

$$\sigma_{\mathcal{Y}_j}(\mathbf{x}) = \max_{\mathbf{y} \in \mathcal{Y}_j} \langle \mathbf{x}, \mathbf{y} \rangle, \quad j = 1, \ldots, c. \tag{3}$$

This enables a two-stage search: first identify promising clusters via the learned scores, then search exhaustively within those clusters. We describe how both SUPPORTNET and KEYNET extend to this multi-output setting below. The single-output case corresponds to $c = 1$.

### 3.1. Model Architectures

**SUPPORTNET: ICNN-based Architecture.** SUPPORT-NET uses an Input Convex Neural Network (ICNN) architecture that guarantees the convexity of each of the $c$ output values of $f_\theta(\mathbf{x})$ w.r.t. **x** (Amos et al., 2017):

$$\mathbf{z}_1 = \sigma(\mathbf{W}_0^{(x)} \mathbf{x} + \mathbf{b}_0) \in \mathbb{R}^h,$$
$$\mathbf{z}_{i+1} = \sigma(\mathbf{W}_i^{(z)} \mathbf{z}_i + \mathbf{W}_i^{(x)} \mathbf{x} + \mathbf{b}_i) \in \mathbb{R}^h,$$
$$f_\theta(\mathbf{x}) = \mathbf{W}_L \mathbf{z}_L + \mathbf{b}_L \in \mathbb{R}^c,$$

where $\mathbf{W}_i^{(z)} \geq 0$ (element-wise non-negative), and $\sigma$ is a convex non-decreasing activation. The "passthrough" connections $\mathbf{W}_i^{(x)} \mathbf{x}$ provide expressiveness while the non-negativity constraint on $\mathbf{W}_i^{(z)}$ ensures convexity is preserved through composition. The output dimension is $c$, corresponding to the number of clusters. For $c = 1$, the network outputs a scalar approximating $\sigma_{\mathcal{Y}}(\mathbf{x})$. For $c > 1$, each output $f_\theta(\mathbf{x})_j$ approximates $\sigma_{\mathcal{Y}_j}(\mathbf{x})$. At inference, we compute the predicted optimal match for cluster $j$ as $\mathbf{y}_{\text{pred},j} = \nabla_{\mathbf{x}} f_\theta(\mathbf{x})_j$ via automatic differentiation. Collectively, the $c$ predicted keys form the rows of the Jacobian matrix $\mathbf{J}_{\mathbf{x}} f_\theta(\mathbf{x}) \in \mathbb{R}^{c \times d}$, where the $j$-th row corresponds to $\nabla_{\mathbf{x}} f_\theta(\mathbf{x})_j$. For $c = 1$, this simplifies to $\mathbf{y}_{\text{pred}} = \nabla_{\mathbf{x}} f_\theta(\mathbf{x})$.

**KEYNET: Direct Key Regression.** While SUPPORTNET elegantly leverages convexity, computing gradients at inference adds computational overhead. We propose KEYNET,

an architecture that directly outputs an approximation of the optimal key without requiring gradient computation:

$$\mathbf{z}_1 = \sigma(\mathbf{W}_0^{(x)} \mathbf{x} + \mathbf{b}_0) \in \mathbb{R}^h,$$
$$\mathbf{z}_{i+1} = \sigma(\mathbf{W}_i^{(z)} \mathbf{z}_i + \mathbf{W}_i^{(x)} \mathbf{x} + \mathbf{b}_i) \in \mathbb{R}^h,$$
$$F_\theta(\mathbf{x}) = \mathbf{W}_L \mathbf{z}_L + \mathbf{b}_L \in \mathbb{R}^{c \times d}.$$

Unlike SUPPORTNET, we enforce no constraint on the parameters of KEYNET. The output is a $c \times d$ matrix (reshaped from the final layer), where each row $F_\theta(\mathbf{x})_j \in \mathbb{R}^d$ is the predicted optimal key for cluster $j$. For $c = 1$, the output is simply a $d$-dimensional vector $F_\theta(\mathbf{x}) \approx \mathbf{y}^\star(\mathbf{x})$. While MLP architectures that are guaranteed to output gradients of convex functions have been proposed (Richter-Powell et al., 2021), their implementation is far too complex at the scales (going as far as hundreds of millions of parameters) we consider in this work. Similarly, we have explored using convex regularizers such as the Monge gap (Uscidda & Cuturi, 2023) but given our unusual setting in which ground-truth pairings are know, this did not yield improvements. We explored two additional modifications to these base architectures.

*Residual blocks.* When enabled, we use ResNet-style residual connections for hidden layers:

$$\mathbf{z}_{i+1} = \mathbf{z}_i + \sigma(\mathbf{W}_i^{(z)} \mathbf{z}_i + \mathbf{W}_i^{(x)} \mathbf{x} + \mathbf{b}_i),$$

when the hidden dimensions of two consecutive states match—this is always the case by default in our setting given the constant hidden dimension. For SUPPORTNET, this modification is convexity-preserving since adding a convex function to the identity (a linear, hence convex, function) yields a convex function. We observe that models using residual blocks take longer to train but can achieve slightly better performance.

*Sparse input injection.* Rather than injecting the input **x** at every layer via $\mathbf{W}_i^{(x)} \mathbf{x}$, we can inject it only at a subset of layers. Let $n_x \in \{0, \ldots, L - 1\}$ denote the number of layers (after the first) where **x** is reinjected. Setting $n_x = 0$ (no injection after the first layer) recovers a standard feedforward architecture.

**Network Sizing.** We use rectangular (constant-width) networks with hidden dimension $h$ across all $L$ hidden layers. We denote the output dimension $d_{\text{out}} = c$ for SUPPORTNET, $c \cdot d$ for KEYNET, and $n_x$ as the number of times the input is reinjected. The total parameter count is $(1 + n_x)dh$ (input projections) $+ (L - 1)h^2$ (hidden-to-hidden) $+ h \cdot d_{\text{out}}$ (output) $+ Lh + d_{\text{out}}$ (biases). Given a total **parameter budget** $P := \rho \cdot n \cdot d$, parameterized as a fraction $\rho \leq 1$ of the database size, the dominant constraint is:

$$(L - 1) \cdot h^2 + (1 + n_x)d \cdot h \approx P. \tag{4}$$

Solving this quadratic yields, writing $D = (1 + n_x)d$:

$$h \approx \frac{\sqrt{D^2 + 4(L-1) \cdot P} - D}{2(L-1)} . \qquad (5)$$

This formula admits two intuitive limiting cases. For *deep networks* ($L$ large), the $h^2$ term dominates and $h \approx \sqrt{P/(L-1)} = \sqrt{\rho \cdot n \cdot d/(L-1)}$—width scales as the square root of budget-per-layer. For *shallow networks with frequent reinjection* ($n_x$ large), the linear term dominates and $h \approx P/D = \rho \cdot n/(1+n_x)$—width scales linearly with budget, divided among reinjection layers.

**Enforcing Homogeneity for SUPPORTNET.** The ground-truth support function $\sigma_\mathcal{Y}(\mathbf{x})$ is positively 1-homogeneous. To enforce this structure in our SUPPORTNET learned models, we have explored two directions: (1) setting all bias vectors $\mathbf{b}_i$ to zero and using a ReLU activation for $\sigma$, which guarantees by construction that property, or (2) relying on a more flexible *homogenization wrapper*: Given any base model $g : \mathbb{R}^d \to \mathbb{R}$, we define:

$$\mathcal{H}[g](\mathbf{x}) := \|\mathbf{x}\| \cdot g\left(\frac{\mathbf{x}}{\|\mathbf{x}\|}\right). \qquad (6)$$

This construction guarantees positive 1-homogeneity regardless of the form of $g$. In practice, we have found this wrapper approach to be more flexible and efficient.

## 3.2. Loss Functions

A crucial feature of amortized MIPS is that we assume access to a *known* relevant distribution of queries $p_\mathcal{X}$, from which we can sample at train time. That distribution *guides* the design of loss functions that optimize model performance for queries drawn from $p_\mathcal{X}$, rather than for arbitrary queries. As a result, all expectations below are taken with respect to $\mathbf{x} \sim p_\mathcal{X}$, and the model is explicitly tailored to achieve a good performance on that distribution of queries.

**Losses for SUPPORTNET.** We combine two complementary losses. Let $\mathbf{y}_j^\star(\mathbf{x}) = \arg\max_{\mathbf{y} \in \mathcal{Y}_j} \langle \mathbf{x}, \mathbf{y} \rangle$ denote the optimal key for cluster $j$ given query $\mathbf{x}$. We first consider a *score* regression loss that ensures that the network output approximates the support function for each cluster:

$$\mathcal{L}_{\text{score}}(\theta) = \mathbb{E}_{\mathbf{x} \sim p_\mathcal{X}} \left[ \frac{1}{c} \sum_{j=1}^{c} \left( f_\theta(\mathbf{x})_j - \sigma_{\mathcal{Y}_j}(\mathbf{x}) \right)^2 \right].$$

We add to that loss a *gradient matching* term that encourages the gradient to match the optimal key for each cluster:

$$\mathcal{L}_{\text{grad}}(\theta) = \mathbb{E}_{\mathbf{x} \sim p_\mathcal{X}} \left[ \frac{1}{c} \sum_{j=1}^{c} \left\| \nabla_\mathbf{x} f_\theta(\mathbf{x})_j - \mathbf{y}_j^\star(\mathbf{x}) \right\|_2^2 \right].$$

Notice that the minimization of the latter term requires computing (using autodiff) the *cross*-derivatives of $f_\theta(\mathbf{x})$ w.r.t $\mathbf{x}$ and then $\theta$. The combined SUPPORTNET objective is a weighted sum of the two.

**Losses for KEYNET.** We focus first on regressing directly the optimal key, and add a consistency constraint: we first encourage that predicted vectors match their targets:

$$\mathcal{L}_{\text{key}}(\theta) = \mathbb{E}_{\mathbf{x} \sim p_\mathcal{X}} \left[ \frac{1}{c} \sum_{j=1}^{c} \left\| F_\theta(\mathbf{x})_j - \mathbf{y}_j^\star(\mathbf{x}) \right\|_2^2 \right],$$

and complement that fitting error with a *score consistency* term: Since KEYNET does not output a score directly, but a vector (or a set of vectors), we derive the predicted score as $\langle F_\theta(\mathbf{x})_j, \mathbf{x} \rangle$ and penalize deviation from the target score:

$$\mathcal{L}_{\text{consist}}(\theta) = \mathbb{E}_{\mathbf{x} \sim p_\mathcal{X}} \left[ \frac{1}{c} \sum_{j=1}^{c} \left( \langle F_\theta(\mathbf{x})_j, \mathbf{x} \rangle - \sigma_{\mathcal{Y}_j}(\mathbf{x}) \right)^2 \right].$$

This consistency loss is motivated by Euler's theorem for homogeneous functions: if $f$ is positively 1-homogeneous, then $\langle \nabla f(\mathbf{x}), \mathbf{x} \rangle = f(\mathbf{x})$. While KEYNET does not explicitly parameterize a grad-potential, this loss encourages the output to behave as if it were the gradient of a 1-homogeneous function.

## 3.3. Training Procedure

Before training, we precompute MIPS solutions for queries sampled from $p_\mathcal{X}$. For the clustered variant with $c$ clusters, we precompute:

$$\mathcal{D} = \left\{ \left( \mathbf{x}_i, \{\mathbf{y}_{i,j}^\star\}_{j=1}^c \right) \right\}_{i=1}^N,$$

from which we can trivially recompute the corresponding scores within each cluster. Here $\mathbf{y}_{i,j}^\star = \arg\max_{\mathbf{y} \in \mathcal{Y}_j} \langle \mathbf{x}_i, \mathbf{y} \rangle$ is the optimal key within cluster $j$ for query $\mathbf{x}_i$. For $c = 1$, this reduces to storing a single optimal key per query. We use exhaustive search for this step.

**Query Augmentation.** Since we have full access to the ground truth generation process, we can easily avoid any limitations on training data through random augmentations of the queries. We have seen that this improves generalization, and we typically augment training queries with Gaussian noise: $\tilde{\mathbf{x}} = \mathbf{x} + \varepsilon$, $\varepsilon \sim \mathcal{N}(0, \sigma^2 \mathbf{I})$. Note that this requires recomputing all targets for each augmented query.

**Activation Function.** We use a soft leaky ReLU activation:

$$\sigma_{\alpha,\beta}(x) = \alpha x + \frac{(1-\alpha)}{\beta} \log(1 + e^{\beta x}),$$

where $\alpha$ is the negative slope and $\beta$ controls smoothness. As $\beta \to \infty$, this approaches the standard leaky ReLU. The

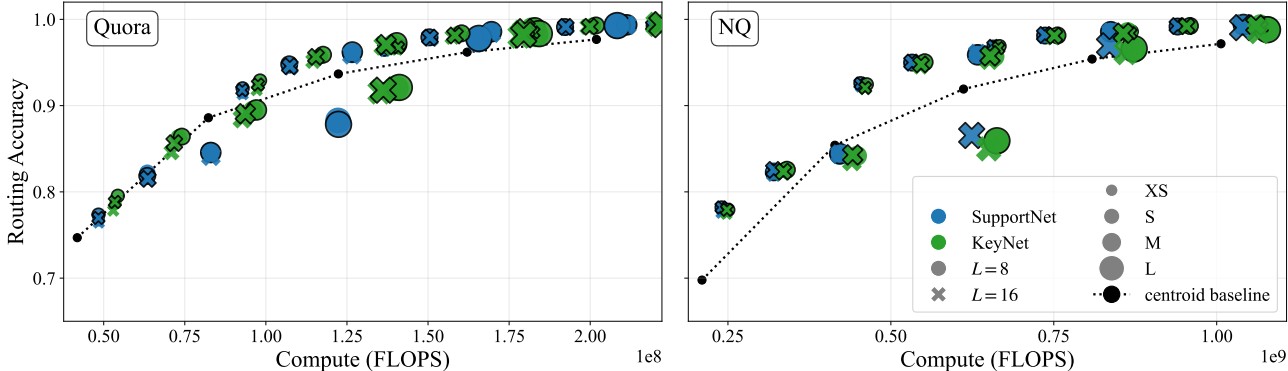

*Figure 3.* Results for the routing experiment described in Section 4.3, on **Quora** (*left*, $n \approx 500k$) and **NQ** (*right*, $n \approx 2.5M$). Here SUPPORTNET and KEYNET are used to predict the support function of a query on $c = 10$ clusters of the entire set of keys $\mathcal{Y}$, followed by exhaustive search within the top scored cluster. For the baseline clustering approach, the cluster is selected first using the top-scoring centroid to the query. These results show that multiple models (trained with various depths $L$, size $\rho$, peak learning rate etc.) achieve consistently a better routing accuracy with a lower FLOPS budget, highlighting the stability of the performance of SUPPORTNET and KEYNET training pipelines to various hyperparameters. Our recommendation, in any practical deployment scenario, would be to select one of those very best performing models depending on the trade-off sought in practice. Markers outlined with a black line correspond to settings in which the input $\mathbf{x}$ is re-injected every 4 layers ($n_x \approx L/4$) whereas no outline indicates reinjection at every layer ($n_x = L$).

fact that the ground-truth support function $\sigma_{\mathcal{Y}}$ is piecewise affine motivates using $\beta \approx \infty$ for SUPPORTNET, but we observe that using $\beta = 20$ strikes a good trade-off between smoothness (better gradient propagation) and adherence to the paradigm. We use $\alpha = 0.1$. While KEYNET should in principle be modeled as a piecewise constant function, we also use the same activation function.

## 4. Experiments

We evaluate our amortized MIPS approach on retrieval benchmarks, comparing SUPPORTNET (score-based with gradient computation) and KEYNET (direct key regression).

### 4.1. Experimental Setup

**Datasets.** We evaluate on four datasets from the BEIR benchmark (Thakur et al., 2021): FIQA, Quora, Natural Questions (NQ), and HotpotQA. These corpora ensure robustness across complementary domains and retrieval styles, ranging from financial QA (FIQA) and semantic duplicate detection (Quora) to factoid extraction from Wikipedia (NQ) and multi-hop reasoning (HotpotQA). Furthermore, they allow us to assess scalability across order-of-magnitude differences in database size, containing approximately **52K**, **513K**, **2.6M**, and **5.2M** keys, respectively; Appendix A.9 additionally reports a scaling study on **BioASQ** ($n \approx 15M$ keys). In all cases, we treat questions as queries and the associated passages or answers as the database keys $\mathcal{Y}$. All text is encoded into $d = 384$ dimensional L2-normalized embeddings using the general-purpose `all-MiniLM-L6-v2` (Reimers & Gurevych, 2019) encoder; Appendix A.5 confirms that our conclusions extend to higher-dimensional encoders (MPNet and Distil-

RoBERTa, $d = 768$) without retuning. We use the encoder in an instruction-free manner, passing raw queries and passages without task-specific conditioning. Queries are short sentences while keys are longer text passages, so the distribution $p_{\mathcal{X}}$ intrinsically differs from that of keys $\mathcal{Y}$; we quantify this shift on each BEIR corpus in Appendix A.10.

**Data Preparation.** Our pre-processing pipeline consists of de-duplication, encoding, and target computation. First, we perform exact string deduplication on both query and document sets. We then encode the unique queries and keys, and augment the resulting *query embeddings* (*not* the keys) by injecting additive Gaussian noise ($\sigma = 0.02$ based on a preliminary study), expanding the query set by a factor of 5–100× depending on the size of the dataset. The augmented embeddings are re-normalized onto the unit sphere and split into train and a validation set of 1000 vectors used for all evaluation metrics. Finally, for every query, we compute the top-$k$ retrieved candidates from the database via exact MIPS. The indices of the maximal inner-product solutions serve as the ground-truth targets for supervision during training. For clustering-based experiments, we further partition the database using $k$-means and compute the optimal keys within each cluster.

**Architecture Configuration & Training.** Both SUPPORTNET and KEYNET use rectangular (constant-width) networks inferred from rule (5) using **Depth** $L \in \{4, 8, 16\}$ hidden layers, **Parameters fraction** $\rho \in \{\text{XS} = 1\%, \text{S} = 5\%, \text{M} = 10\%, \text{L} = 20\%, \text{XL} = 40\%, \text{XXL} = 50\%\}$ of $n \times d$ database size, and **Passthrough** $n_x$ total input skip connections. We always apply the Homogenize wrapper to SUPPORTNET to enforce positive 1-homogeneity. We train models with Adam using a cosine learning rate schedule

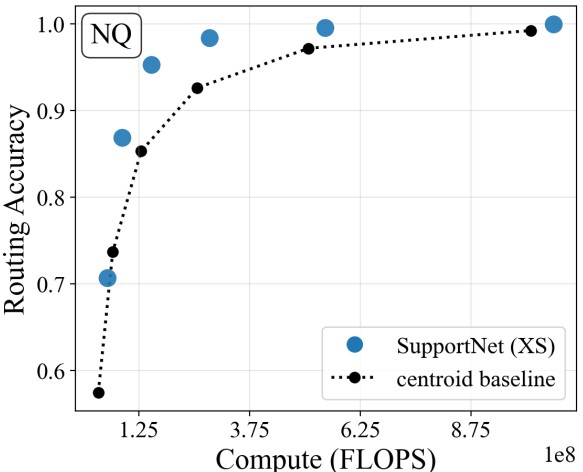

*Figure 4.* Routing accuracy vs. compute (FLOPs) on **NQ** with $c = 128$ clusters, for an XS SUPPORTNET ($L = 8$, 1B training samples) and the centroid baseline. Each point corresponds to exhaustive within-cluster search across the top-$k$ buckets, with $k \in \{1, 2, 4, 8, 16, 32\}$.

with warmup (2.5% of training horizon) and peak value at $\approx 10^{-4}$. For both models, we emphasize the key reconstruction as training signal: we use $\lambda_{\text{score}} = 0.01$ and $\lambda_{\text{grad}} = 1.0$ for SUPPORTNET; $\lambda_{\text{key}} = 1.0$ and $\lambda_{\text{consist}} = 0.01$ for KEYNET (Appendix A.6 reports the ablation supporting these defaults). We track the EMA of model parameters (decay 0.999 for batch size 128) and use EMA parameters for evaluations. Batch sizes range from 512 to 16384 depending on the dataset size. We rescale the learning rate using the square-root ratio to reference batch size, and scale the EMA decay geometrically with batch-size (Busbridge et al., 2023). We train typically on 1–10B samples; Appendix A.7 shows that downstream metrics plateau past $\approx 3$B samples. See Table 1 in appendix for the time needed to recover both support and key from a query for these networks.

## 4.2. Metrics

Apart from regression losses (to score/key, respectively) and matching terms (gradient/consistency), we evaluate both SUPPORTNET and KEYNET models using two metrics:

**Relative Transport Error.** Since our goal is to learn a transport map $\mathbf{x} \mapsto \mathbf{y}^\star(\mathbf{x})$, we introduce a *relative* error metric (used only for evaluation, not as a training loss) that measures prediction quality relative to the baseline distance between query and target. We write here $\hat{\mathbf{y}}(\mathbf{x})$ for the predicted key of our models, either $\nabla_{\mathbf{x}} f_\theta(\mathbf{x})$ for SUPPORTNET and $F_\theta(\mathbf{x})$ for KEYNET, and define

$$\mathcal{E}_{\text{rel}} = \left( \mathbb{E}_{\mathbf{x}} \left[ \log \frac{\|\hat{\mathbf{y}}(\mathbf{x}) - \mathbf{y}^\star\|_2^2}{\|\mathbf{x} - \mathbf{y}^\star\|_2^2} \right] \right). \quad (7)$$

When the number of clusters $c > 1$, that metric is computed per key-in-cluster prediction, and averaged. This log-ratio measures how much closer (negative values) or farther (positive values) the prediction is to the target compared to the query itself. A value of $\mathcal{E}_{\text{rel}} = 0$ means the prediction error equals the query-to-target squared-distance; $\mathcal{E}_{\text{rel}} = -1$ means the prediction is $e^{-1} \approx 0.37\times$ as far from the target as the query is; a perfect prediction yields $\mathcal{E}_{\text{rel}} \to -\infty$. We observe that our models start performing well when $\mathcal{E}_{\text{rel}}$ goes below $-1$. The training dynamics for that metric are shown in Figure 9.

**Retrieval Metrics.** We track whether the location of the predicted key $\hat{\mathbf{y}}(\mathbf{x})$ agrees, relative to all other keys, with the original query. The *match rate* is the fraction of queries where $\arg\min_{\mathbf{y} \in \mathcal{Y}} \|\hat{\mathbf{y}} - \mathbf{y}\| = \mathbf{y}^\star$. *Recall@k* is the fraction of queries where $\mathbf{y}^\star$ is among the $k$ nearest neighbors of $\hat{\mathbf{y}}$. The *mean reciprocal rank* (MRR) is the mean reciprocal rank of $\mathbf{y}^\star$ in the ranking induced by distance from $\hat{\mathbf{y}}$.

Notice that the relative transport error and retrieval metrics can agree jointly when the prediction is perfect, but do not necessarily agree in other cases. For instance, if the predictor $\hat{\mathbf{y}}$ were simply initialized to be the identity, the RTE would be equal to 1, whereas matching metrics would give the best possible values, by definition. During our training, we have therefore tracked models that are able to transport significantly close to target (RTE $\approx 0$) while maintaining good metrics. The tradeoffs between $\mathcal{E}_{\text{rel}}$ and MRR across model sizes and depths at the end of training are reported in Appendix A.4.

## 4.3. Performance as Routing Mechanism

We evaluate the clustered variant ($c > 1$) where the model simultaneously learns support functions for multiple database partitions. This enables a two-stage search strategy: use the learned scores to identify the top-$k$ most promising clusters, then perform exact search within those clusters.

**Setup.** We begin by clustering the Quora and NQ datasets into $c = 10$ clusters using k-means, with centroids $\{\mathbf{z}_i\}_{i=1}^c$. We run clustering end-to-end 10 times and select the clustering which yields the most even cluster sizes, in order to balance the cost of exact search within each cluster. We then train SUPPORTNET and KEYNET models using $c = 10$, following the conventions in Section 4.1. To obtain a Pareto frontier, we vary the number of clusters $k$ searched at test time, going from only the top-1 cluster up to top-5. For the baseline, we perform routing by comparing the query $\mathbf{q}$ against the cluster centroids $\mathbf{z}_i$; that is, we predict $\arg\max_i \langle \mathbf{q}, \mathbf{z}_i \rangle$. We then evaluate each method in terms of *routing accuracy* (whether the target key is in the selected clusters) versus *compute* (FLOPs needed to perform cluster selection and exact search within the selected clusters).

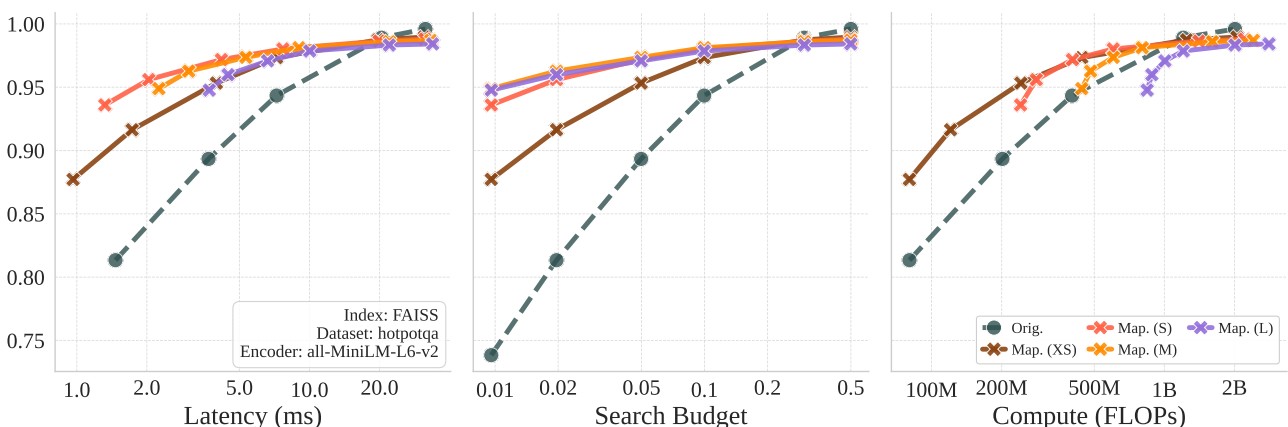

*Figure 5.* FAISS-IVF integration with KEYNET on **HotpotQA** ($n \approx 5.2$M keys), showing Recall@0.1% for XS, S, M, and L KEYNET variants vs. the unmodified query (Orig.), across three cost axes: wall-clock latency (*left*), search budget ($n_{probe}$, *center*), and FLOPs (*right*). Appendix A.8 reports Recall@$\{0.01\%, 0.5\%\}$ and extends to other datasets and backends.

**Results.** On **NQ**, both SUPPORTNET and KEYNET consistently beat the baseline at all budgets, yielding a $\geq 10$ point increase at $k = 1$ with minimal overhead and saturating close to 100% routing accuracy at $k = 5$; larger models (M, L) dominate on this dataset. On the smaller **Quora** set, the baseline performs favorably in the low-cost regime, but SUPPORTNET and KEYNET scale better with increased compute; smaller models (XS, S) dominate here. Both models perform similarly at most sizes, with slight gains for KEYNET; on Quora, SUPPORTNET becomes competitive at limited compute because KEYNET's larger output layer ($h \times cd$ vs. $h \times c$) raises its FLOPs cost. Dense re-injections ($n_x$ close to $L$) and increasing depth $L$ have little impact on performance but may affect training stability.

**Scaling to more clusters.** The $c = 10$ setup above is deliberately conservative in cluster count. To assess whether the routing advantage persists higher cluster counts, we re-run the study on **NQ** with $c = 128$, training an XS SUPPORTNET with $L = 8$ for 1B samples, and compare against the centroid baseline as a function of FLOPs (top-$k$ buckets, with $k \in \{1, 2, 4, 8, 16, 32\}$). In that setup, the output dimension of KEYNET would be too large to handle, which illustrates why we advocate using SUPPORTNET for cluster routing. The learned SUPPORTNET router dominates across the entire low-FLOPs regime (Figure 4): at $k = 1$ it routes correctly $\approx 72\%$ of the time vs. $\approx 56\%$ for the centroid baseline; at $k = 4$ it reaches $\approx 95\%$, matching what the centroid baseline only achieves at $k = 16$ (i.e., at roughly $4\times$ the FLOPs). The qualitative picture from $c = 10$ thus extends to this higher-$c$ regime: the learned router's advantage is largest when the operating budget is tight.

### 4.4. Approximate Search Integration

In this section, we focus on compressing a single dataset of keys, i.e. $c = 1$, and focus on top key retrieval. Because SUPPORTNET requires a backward pass to output that prediction, its FLOPs overhead (starting from the same total parameter count) compared to KEYNET is prohibitive (the backward pass typically costs $1\sim2$ times more than the forward). As a result, we focus on KEYNET in this section.

**Setup.** We feed KEYNET's prediction $\hat{\mathbf{y}}(\mathbf{x})$ as a drop-in replacement for the query vector into a standard FAISS Inverted File (IVF) index (Johnson et al., 2019). The index itself is left unchanged: it partitions $\mathcal{Y}$ into Voronoi cells via a coarse $k$-means quantization of the embedding space, and at query time it (i) selects the $n_{probe}$ cells whose centroids score highest against the input vector, then (ii) exhaustively scans those cells. Because $\hat{\mathbf{y}}(\mathbf{x})$ is, by construction, closer to the true top key than $\mathbf{x}$, handing the index $\hat{\mathbf{y}}(\mathbf{x})$ in place of $\mathbf{x}$ makes step (i) more accurate without any modification to the indexing pipeline. We compare two retrieval strategies under a common $n_{probe}$ sweep: **Original**, which queries the index with the unmodified $\mathbf{x}$, and **Mapped**, which queries it with $\hat{\mathbf{y}}(\mathbf{x})$. Cost is reported across three axes: wall-clock latency (ms), search budget ($n_{probe}$), and FLOPs. We trace Pareto fronts of Recall@$k$ vs. each cost axis, with $k$ expressed as a fraction of $|\mathcal{Y}|$ (Recall@0.1%). Appendix A.8 additionally reports Recall@$\{0.01\%, 0.5\%\}$ and replicates the search integration with other indexers such as ScaNN (Guo et al., 2020), SOAR (Sun et al., 2023), and LeanVec (Tepper et al., 2023), across all datasets.

**Results.** As shown in Figure 5, mapping queries to their predicted keys via KEYNET consistently improves retrieval performance over directly querying the IVF index with the original query vector, across a wide range of compute bud-

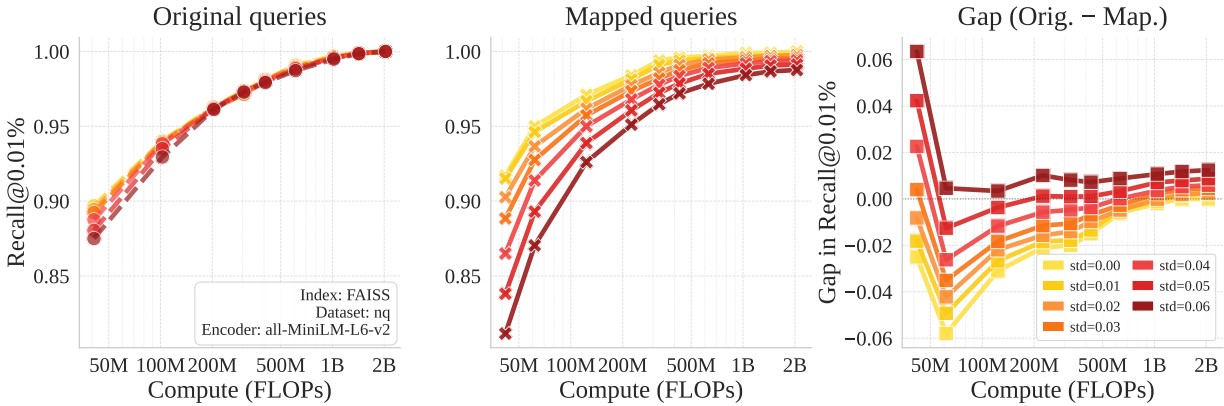

*Figure 6.* Robustness of the FAISS-IVF integration to query distribution shift on **NQ**. Each panel shows Recall@0.01% as a function of compute (FLOPs): original queries (*left*), mapped queries $\hat{\mathbf{y}}(\mathbf{x})$ from an XS KEYNET (*center*), and their gap (*right*; values below zero mean our approach outperforms feeding $\mathbf{x}$ directly to FAISS). Colors encode the noise standard deviation $\sigma \in \{0.00, 0.01, 0.02, 0.03, 0.04, 0.05, 0.06\}$ applied to test queries.

gets. This benefit is particularly pronounced for smaller model sizes (XS, S), which offer a favorable trade-off between the forward-pass overhead and the resulting search improvement: larger models (L) incur a substantially higher base FLOPs cost that does not always translate into commensurate recall gains. The latency plot shows that L models recover competitiveness in wall-clock time due to leverage of parallelism on GPU hardware and efficient batching. Under latency constraints, one might prefer larger KEYNET variants, compute constrained deployment environments should rather favor XS/S. The improvement is consistent across all three cost axes shown in Figure 5.

### 4.5. Robustness to Query Distribution Shift

A practical concern for learned retrieval models is how performance degrades when the query distribution at inference time deviates from the training distribution. We evaluate this by re-running the FAISS-IVF integration with XS KEYNET on **NQ** under progressively noisier test queries: we inject additive Gaussian noise ($\sigma \in \{0.00, 0.01, 0.02, 0.03, 0.04, 0.05, 0.06\}$, larger than the $\sigma = 0.02$ used during training) and re-normalize, creating controlled distribution shifts. Figure 6 reports Recall@0.01% as a function of FLOPs for original queries, mapped queries, and their gap (original−mapped; negative means our approach outperforms baseline). As expected, mapping degrades under distribution shift. Importantly, the degradation is not catastrophic: the improvement over directly querying the index with $\mathbf{x}$ persists through the mid-budget regime even at $\sigma = 0.03$, with the gap most pronounced at the extrema of the search-cost range. This suggests the method retains practical value under moderate query drift. Appendix A.2 reports latency results and corresponding results on **Quora**.

## 5. Conclusion

We have introduced *amortized MIPS*, a learning-based paradigm that trains neural networks to directly predict MIPS solutions on a distribution of queries that matter to the end-user, amortizing the cost of repeated search. Our approach exploits the mathematical structure of MIPS: the value function is the *support function* of the database, a convex, positively 1-homogeneous function whose gradient equals the optimal key. We propose two architectures: SUP-PORTNET learns this convex potential using relaxed ICNNs and recovers matches via automatically differentiated gradients; KEYNET directly regresses the optimal key, bypassing gradients entirely at inference. Both models cover the two settings we studied: cluster routing, and projecting queries toward their optimal keys to ease approximate search.

Our experiments on the BEIR benchmark demonstrate consistent and robust empirical gains. As a cluster router, SUP-PORTNET outperforms centroid-based routing. KEYNET, when combined with various search indexers, consistently improves recall-vs-compute Pareto fronts at scales up to 15M keys, with gains remaining robust to encoder family and moderate query distribution shift. Across both deployment modes, smaller models (XS, S) offer the best compute-recall trade-off, while larger variants recover competitiveness under latency constraints, suggesting that model-size selection should be guided by the deployment cost metric.

**Limitations and future work.** Our approach requires a representative query distribution; performance may degrade under distribution shift (§4.5, App. A.2). Scaling to billion-scale databases requires more efficient pre-computation pipelines that will likely involve the joint use of SUPPORT-NET (as a routing mechanism) before running KEYNET within each cluster. Future work includes online adaptation to query drift, distillation from larger models.

## Impact Statement

This paper presents work whose goal is to advance the field of Machine Learning. There are many potential societal consequences of our work, none which we feel must be specifically highlighted here.

## Contribution Statement

All authors contributed to idea generation, code and paper writing. MC proposed to frame MIPS as a convex function regression problem following discussions on MIPS with MK and JM. JM designed the train (dataloaders) and validation metrics, establishing the overall pipeline with an MLP. MK scaled it up, both for dataloaders and models, adding ICNN, monitoring code health throughout the project. MC proposed adaptive sizing of networks, ICNN variants and the switch to KeyNet models. TXO oversaw the extension to the multi-cluster case, running evaluations for S.4.3. MC monitored training runs for S.4.4. JM proposed and ran the evaluation pipeline for section S.4.4.

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

# A. Appendix

## A.1. Timing Results

In Table 1 we compare the time to compute scores and gradient using SUPPORTNET and KEYNET across dataset sizes and model parameter fractions. For gradient computation, SUPPORTNET requires backpropagation through the network, while KEYNET directly predicts the gradients, resulting in gradient times that closely match the score times.

| Dataset | Parameter Fraction | SUPPORTNET | | KEYNET | |
|---|---|---|---|---|---|
| | | Score | Grad | Score | Grad |
| Quora (513K) | 0.05 (S) | $1.34 \times 10^{-3}$ | $2.50 \times 10^{-3}$ | $1.28 \times 10^{-3}$ | $1.29 \times 10^{-3}$ |
| | 0.1 (M) | $2.18 \times 10^{-3}$ | $4.08 \times 10^{-3}$ | $2.13 \times 10^{-3}$ | $2.12 \times 10^{-3}$ |
| | 0.2 (L) | $1.99 \times 10^{-3}$ | $3.12 \times 10^{-3}$ | $1.90 \times 10^{-3}$ | $1.90 \times 10^{-3}$ |
| Natural Questions (2.6M) | 0.05 (S) | $4.56 \times 10^{-3}$ | $8.64 \times 10^{-3}$ | $4.46 \times 10^{-3}$ | $4.46 \times 10^{-3}$ |
| | 0.1 (M) | $8.55 \times 10^{-3}$ | $1.62 \times 10^{-2}$ | $7.68 \times 10^{-3}$ | $7.69 \times 10^{-3}$ |
| | 0.2 (L) | $7.08 \times 10^{-3}$ | $1.23 \times 10^{-2}$ | $6.00 \times 10^{-3}$ | $5.99 \times 10^{-3}$ |
| HotpotQA (5.2M) | 0.05 (S) | $8.69 \times 10^{-3}$ | $1.68 \times 10^{-2}$ | $8.15 \times 10^{-3}$ | $8.15 \times 10^{-3}$ |
| | 0.1 (M) | $1.59 \times 10^{-2}$ | $3.05 \times 10^{-2}$ | $1.49 \times 10^{-2}$ | $1.49 \times 10^{-2}$ |
| | 0.2 (L) | $3.14 \times 10^{-2}$ | $5.96 \times 10^{-2}$ | $2.85 \times 10^{-2}$ | $2.86 \times 10^{-2}$ |

*Table 1.* Timing comparison between SUPPORTNET and KEYNET across datasets and parameter fractions. *Score* and *Grad* columns show the time in seconds required to process a batch of 4096 queries in dimension 384, averaged over 100 runs. Number of keys for each dataset is shown in parentheses.

## A.2. Robustness to out-of-distribution queries

We test XS variants of KEYNET on perturbed versions of the test sets of **NQ** and **Quora**. Perturbations follow the scheme discussed in Section 4.1, used for augmenting training data; we apply Gaussian noise and re-normalize query representations. In this case however, we made it so noise distributions differ relative to those used to prepare training data, so there are significant shifts on the test query distribution with respect to the source of training queries.

We re-run the FAISS-IVF integration of Section 4.4 under these perturbed queries and report Recall@0.01% as a function of both compute (FLOPs) and wall-clock latency (ms). Figures 7 and 8 track the evolution of recall as the Gaussian noise standard deviation $\sigma$ grows over $\{0.00, 0.01, 0.02, 0.03, 0.04, 0.05, 0.06\}$. As one would expect, learning-based approaches degrade under distribution shift. Interestingly, however, the degradation is not catastrophic: models still achieve relatively high recall in absolute terms even at noise levels well above what was observed during training. The gap with respect to original queries (original − mapped) is most pronounced at the extrema of the search-cost budget; through the middle of the operating range the mapping continues to outperform feeding $\mathbf{x}$ directly to FAISS even at $\sigma = 0.03$. This residual degradation can likely be alleviated by more aggressive data augmentation, though augmentation alone would not render the model robust to other types of distribution shift.

## A.3. Training Dynamics

We track the relative transport error $\mathcal{E}_{\text{rel}}$ (Section 4) along the course of training to verify that the optimization is stable across the model sizes and depths we sweep over, and to expose the marginal benefit of additional capacity. Figure 9 reports these trajectories on **Quora**; the curves separate cleanly by size, with no sign of divergence or late-training instability, and larger models keep improving for longer before plateauing.

## A.4. Training Trade-offs: MRR vs. $\mathcal{E}_{\text{rel}}$

Figure 10 shows the transport ($\mathcal{E}_{\text{rel}}$) and retrieval (MRR) errors achieved by our models on FIQA (50k) and QUORA (500k), across different depths and sizes. Model size appears to be the most important driver of increased performance, consistently improving both metrics for both SUPPORTNET and KEYNET. Furthermore, shallower networks generally do better than their

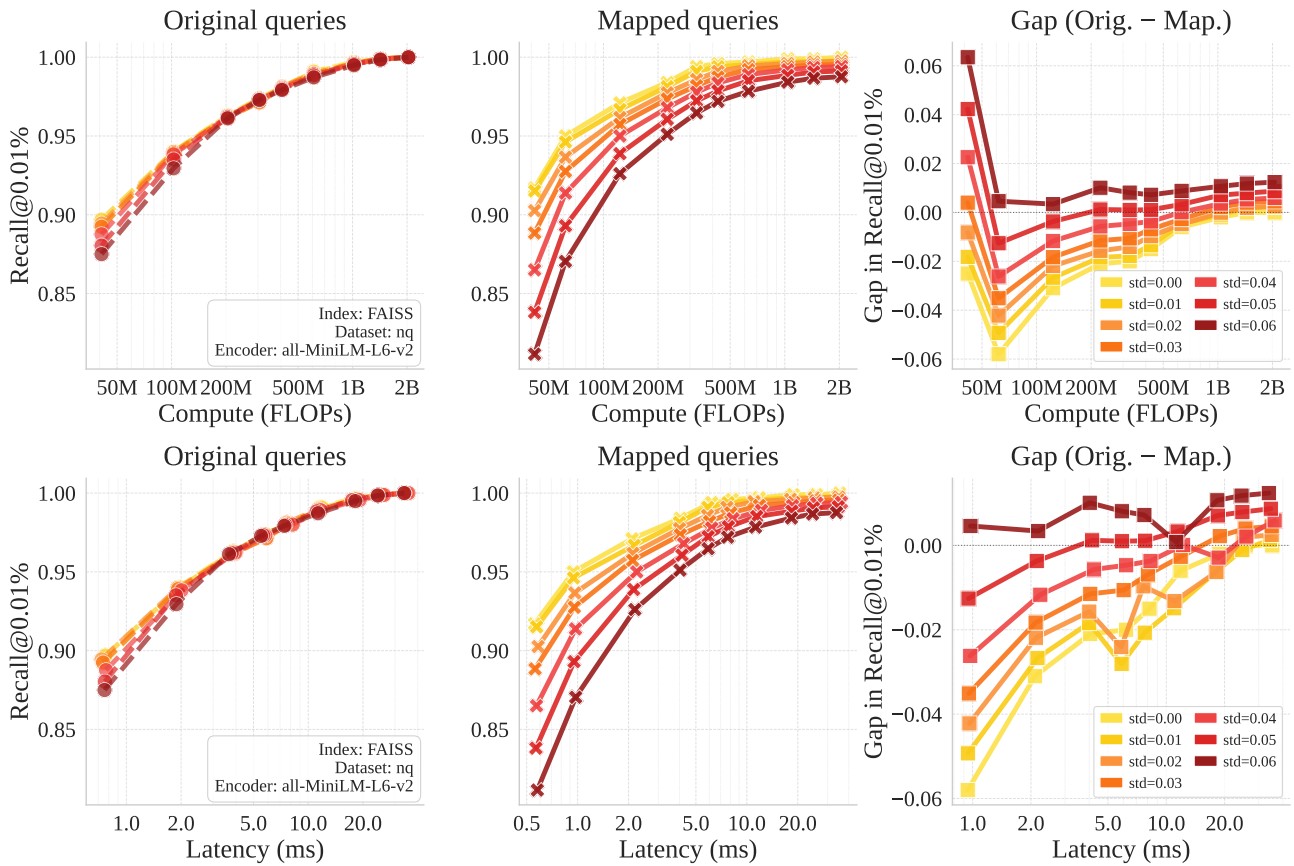

*Figure 7.* Evolution of Recall@$0.01\%$ under increasing distribution shift in the query test distribution on **NQ**, i.e., under growing Gaussian noise standard deviation $\sigma \in \{0.00, 0.01, 0.02, 0.03, 0.04, 0.05, 0.06\}$ (colors). Plots are for an XS KEYNET feeding a FAISS-IVF index (Section 4.4). Each row shows three panels: recall of the *original* queries (*left*), of the *mapped* queries $\hat{\mathbf{y}}(\mathbf{x})$ (*center*), and the gap original$-$mapped (*right*; lower than 0 means our approach outperforms feeding $\mathbf{x}$ directly to FAISS). Cost is reported as compute (FLOPs, *top row*) and wall-clock latency (ms, *bottom row*). The gap is most pronounced at the extrema of the search-cost budget.

deeper counterparts, particularly at larger model sizes. As to differences between SUPPORTNET and KEYNET, both models are capable of achieving low transport error and high MRR on FIQA. However, as model size increases, SUPPORTNET appears to trade off a lower transport error for slightly lower MRR, while the opposite is true at lower budgets.

### A.5. Scaling to higher-dimensional encoders ($d = 768$)

The experiments in Section 4 use `all-MiniLM-L6-v2` ($d = 384$). To assess whether KEYNET scales to higher-dimensional representations, we re-train it out of the box on **Quora** and **NQ** with two $d = 768$ encoders: `all-mpnet-base-v2` (MPNet) and `all-distilroberta-v1` (DistilRoBERTa). All hyperparameters are kept identical to the MiniLM runs; only the embedding dimension changes. Per the sizing rule of Eq. (5), the parameter budget at a fixed fraction $\rho$ scales linearly with $d$, not quadratically: the hidden width $h$ is determined jointly by $\rho$, $n$, $L$, and $d$, and the network's $h \times h$ MLP layers do not blow up with the input dimension.

Before evaluating downstream retrieval, we first check that training behaves as expected at $d = 768$. Figure 11 plots the log gradient error $\mathcal{E}_{\text{rel}}$ versus training samples seen for the XS (pf=1%) and S (pf=5%) KEYNET variants on top of both encoders. Curves decrease smoothly throughout training on both datasets and reach values comparable to those of the MiniLM runs in Figure 9; larger S variants reach lower asymptotic error than XS, but both encoders behave near-identically at each capacity. This confirms that the training recipe transfers to $d = 768$ out of the box.

Figures 12 and 13 report Recall@$\{0.01\%, 0.1\%, 0.5\%\}$ for the FAISS-IVF integration of Section 4.4, against both compute (FLOPs) and wall-clock latency (ms), for XS and S KEYNET variants on top of each encoder. Mapping queries with KEYNET dominates direct retrieval over the mid-budget regime in all four (dataset, encoder) combinations, mirroring the

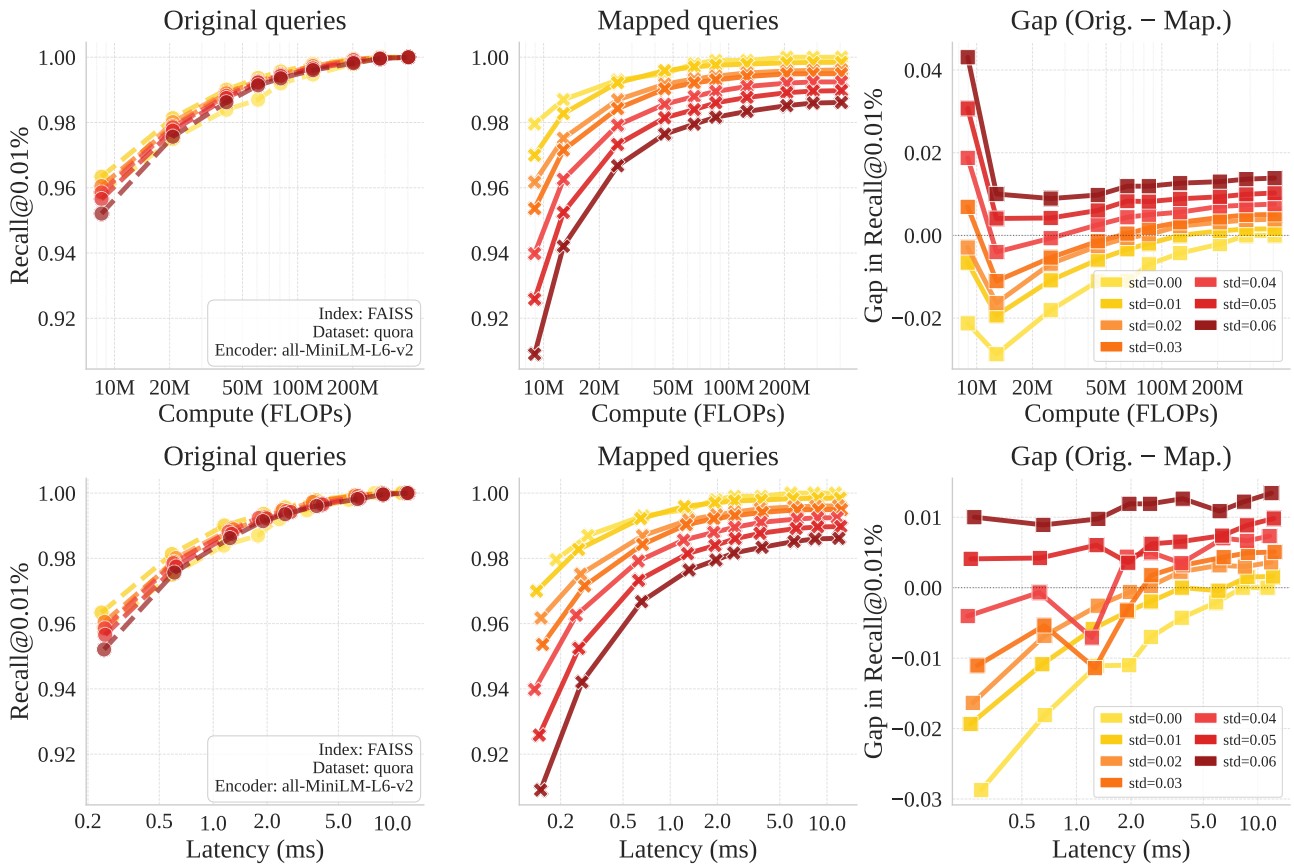

*Figure 8.* Same setting as Figure 7, on **Quora**. The gap to feeding **x** directly to FAISS narrows here relative to NQ, in line with the smaller query/key distribution shift observed for Quora (see Appendix A.10, Figure 29).

behavior reported for MiniLM in Section 4.4. The improvement is robust to the choice of encoder, indicating that the amortization signal does not depend on a particular sentence-encoder family or embedding size.

### A.6. Loss-weight ablation

The training objectives for SUPPORTNET and KEYNET combine two complementary terms (score regression with gradient matching for SUPPORTNET; key regression with score consistency for KEYNET). The defaults reported in Section 4 ($\lambda_{\text{score}} = 0.01$, $\lambda_{\text{grad}} = 1.0$ for SUPPORTNET; $\lambda_{\text{key}} = 1.0$, $\lambda_{\text{consist}} = 0.01$ for KEYNET) deliberately emphasize key/gradient reconstruction; this appendix shows what each term buys when isolated.

Figure 14 plots gradient error versus score error on NQ with MiniLM, $L = 16$, for three loss configurations (grads-only with $\lambda_{\text{grad}} = 1$; scores-only with $\lambda_{\text{score}} = 10^{-2}$; the combined default), each run with two peak learning rates and across both SUPPORTNET and KEYNET.

The two single-objective configurations land in opposite corners as expected: scores-only (blue) achieves the lowest score error but the highest gradient error, while grads-only (red) does the reverse. The combined loss (green) sits much closer to the grads-only point on the y-axis than to the scores-only point on the x-axis, confirming that a small score weight ($10^{-2}$ relative to the gradient weight) reduces score error noticeably without measurably degrading gradient error. This is the regime our defaults sit in for both models.

### A.7. Training horizon vs. performance trade-offs

How long should we train? We sweep the training horizon over $\{1, 3, 5, 7\}$ billion samples seen for a pf=5% (S) KEYNET model with $L = 16$ on **NQ**/MiniLM, with all other hyperparameters held fixed. Figure 15 reports both the optimization

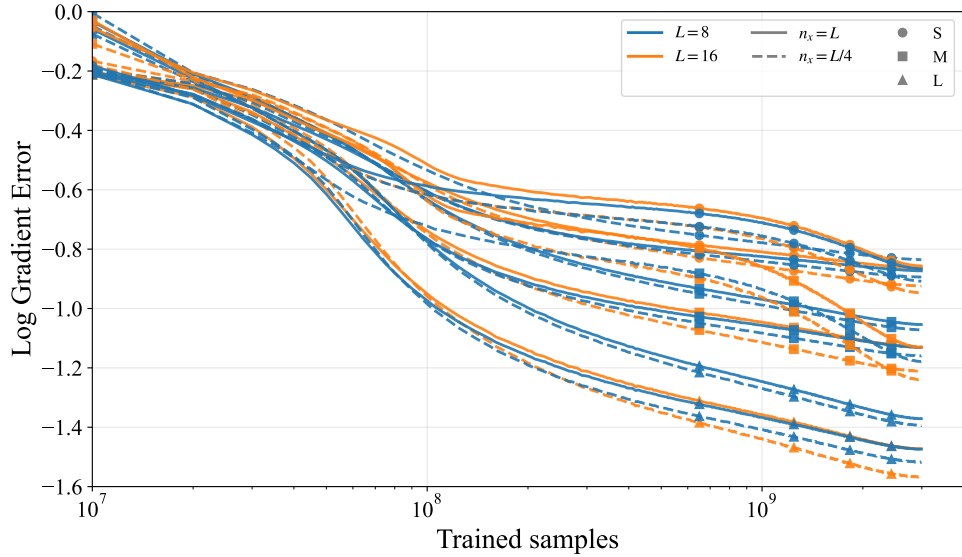

*Figure 9.* Evolution of the relative transport error $\mathcal{E}_{\text{rel}}$ during training on **Quora** for KEYNET across different model sizes. Lower values indicate predictions closer to the ground-truth key. Larger models achieve lower final error and favor greater depth.

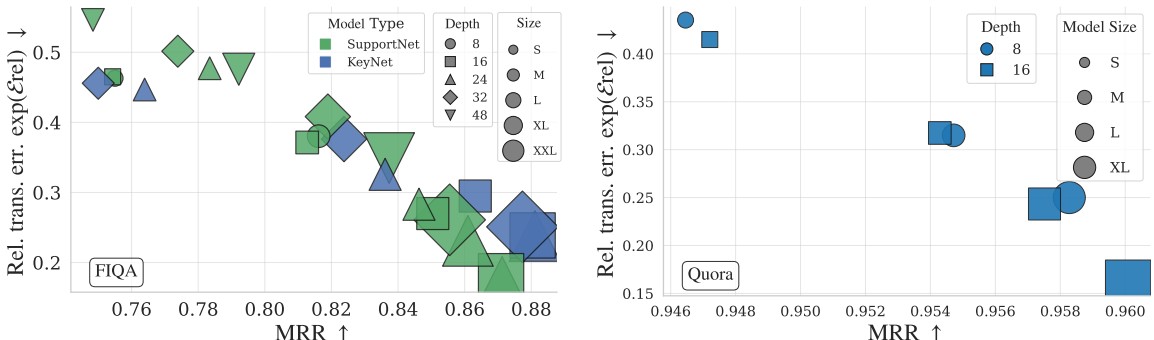

*Figure 10.* $\mathcal{E}_{\text{rel}}$ vs. MRR metric on various model sizes and depths for SUPPORTNET and KEYNET at the end of training on FIQA (*left*) and QUORA (*right*, only KEYNET reported on that task as used later in Section 4.4). Lower right corner (i.e., high MRR and low $\mathcal{E}_{\text{rel}}$) is best.

trace (training loss vs. samples seen, *left*) and the downstream Pareto position (relative transport error vs. MRR, *right*) at the end of each horizon.

The training loss keeps decreasing through all four horizons, with longer runs reaching lower asymptotic values. The downstream metrics, however, show diminishing returns past roughly 3B samples: $\exp(\mathcal{E}_{\text{rel}})$ drops from $0.40$ at 1B to $0.375$ at 3B but only to $0.365$ by 7B, and MRR is essentially saturated between 3B and 7B (gain of $\approx 0.002$). In other words, additional training continues to improve the optimization objective long after the metrics practitioners care about have stopped moving meaningfully. We therefore treat $\approx$ 3B samples as the practical sweet spot for the S model on NQ.

### A.8. Extended cost metrics and additional indexing backbones

The integration evaluation in Section 4.4 reports Recall@0.1% on **HotpotQA** against three cost axes. This appendix extends to all three BEIR datasets (Quora, NQ, HotpotQA), to Recall@$\{0.01\%, 0.1\%, 0.5\%\}$, and replicates the integration with three further indexing backbones: **ScaNN** (Guo et al., 2020), **SOAR** (Sun et al., 2023), and **LeanVec** (Tepper et al., 2023). These backbones span complementary design choices: anisotropic quantization (ScaNN), redundant spilled assignments (SOAR), and learned linear vector compression (LeanVec).

Across all four backbones and all three cost axes, the qualitative picture is unchanged: mapping queries with KEYNET

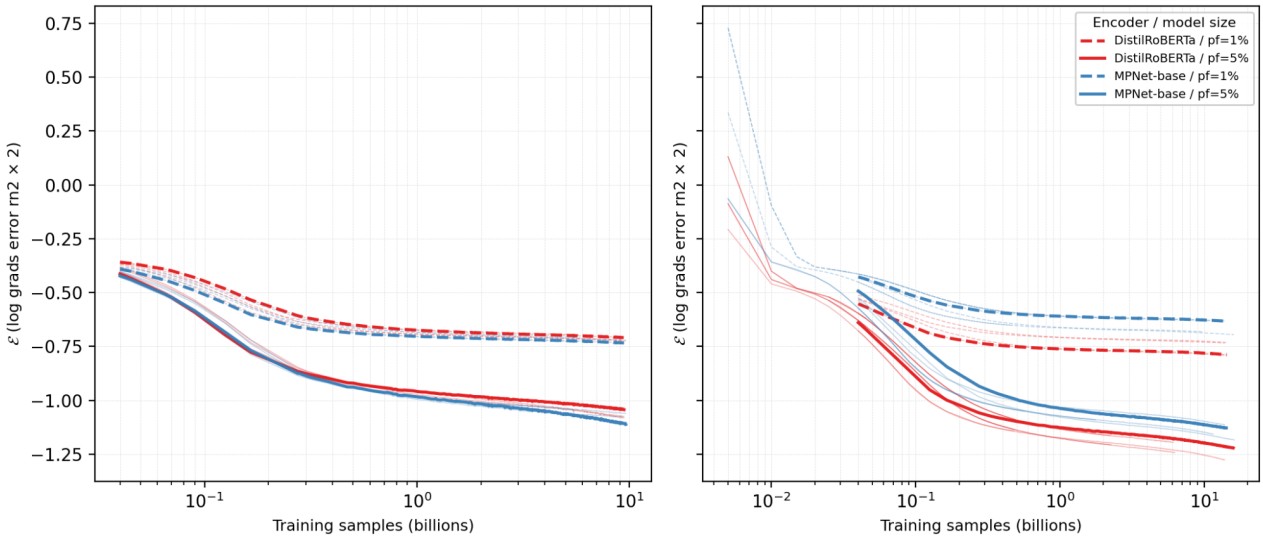

*Figure 11.* Training curves for KEYNET on top of two $d = 768$ encoders, MPNet and DistilRoBERTa, at pf=1% (XS) and pf=5% (S). Log gradient error $\mathcal{E}_{\text{rel}}$ vs. training samples (billions) on **NQ** (*left*) and **Quora** (*right*). All hyperparameters are kept identical to the MiniLM runs of Figure 9.

dominates direct retrieval in the mid-budget regime, with the largest improvements at low XS/S sizes; only at very low budgets, where the forward-pass overhead is not yet amortized by the index speed-up, does direct retrieval remain competitive. Crossover budget and gap size depend on the backbone, but the ordering between *Original* and *Mapped* curves is stable across datasets, axes, and indexers.

**FAISS-IVF.** Figures 16, 17, and 18 report the FAISS-IVF integration on Quora, NQ, and HotpotQA across all three cost axes and Recall@$\{0.01\%, 0.1\%, 0.5\%\}$.

**ScaNN.** We replace the FAISS-IVF index with ScaNN (Guo et al., 2020), which uses anisotropic vector quantization to bias quantization error along directions less likely to harm inner-product estimates. This is the most direct learned-quantization baseline highlighted in prior work as already using a notion of data distribution at index time. Figures 19–21 show that, despite the stronger baseline, KEYNET-mapped queries still improve over original queries through the mid-budget regime; gains shrink at very high budgets where ScaNN is near saturation.

**SOAR.** SOAR (Sun et al., 2023) extends ScaNN's IVF structure by assigning each database point to multiple redundant cells, so that a near-orthogonal residual to the primary cell centroid is captured by a secondary one. Figures 22–24 report results. SOAR is the most aggressive backbone in our set: it converges quickly to high recall at moderate budgets, leaving a narrower regime where KEYNET can help. On Quora the index is essentially saturated and the curves overlap; on NQ and HotpotQA the mid-budget improvement persists.

**LeanVec.** LeanVec (Tepper et al., 2023) compresses the database with a learned linear projection that minimizes inner-product distortion before indexing, complementing rather than replacing the IVF structure. Figures 25–27 report results. The pattern mirrors the FAISS case, with a slight latency advantage on small budgets owing to the compressed representation.

### A.9. Scaling the database to 15M keys (BioASQ)

The largest BEIR corpus used in the main paper is **HotpotQA** ($n \approx 5.2$M). To probe whether the amortized mapping continues to pay off at substantially larger scale, we run an additional FAISS-IVF integration on **BioASQ** (Tsatsaronis et al., 2015), a biomedical question-answering corpus with $n \approx 15$M keys, roughly 3× HotpotQA and 6× NQ. Encoder (all-MiniLM-L6-v2, $d = 384$), training pipeline, and FAISS-IVF configuration are kept identical to Section 4.4; only the database changes.

Figure 28 reports the result. The picture established at smaller scales extends cleanly: mapping queries through KEYNET dominates direct retrieval across the mid-budget regime under all three cost axes (FLOPs, search budget, latency); the XS and S models again offer the best amortized trade-off, with the larger M variant only catching up at the higher end of the budget where the forward-pass overhead is fully repaid by index speed-up. The absolute Recall curves shift downward relative to smaller corpora, as one would expect from a larger candidate pool, but the relative gap between original and mapped queries does not collapse.

### A.10. Distributional differences between queries and keys

The amortization argument in Section 4 relies on the queries and keys being drawn from *distinct* distributions: if $p_\mathcal{X}$ and $p_\mathcal{Y}$ coincide, then any query already looks like a plausible key and mapping queries with KEYNET buys little. Conversely, the larger the shift, the more room a learned map has to move a query toward the manifold of keys. This appendix provides direct visual evidence that the BEIR corpora we use exhibit such a shift, and that its magnitude is consistent with where our method shines.

**Joint and marginal density estimates.** Figure 29 shows, for each of **Quora**, **NQ**, and **HotpotQA**, a 2D projection of the embeddings (PCA into the leading two principal components of the keys), together with separate kernel density estimates for keys and queries. Note the color-bar scales: keys and queries are normalized independently, so the panels are best compared by the *shape* and *location* of their high-density regions rather than absolute height. On NQ and HotpotQA the query distribution is visibly displaced relative to the keys, with several query-side modes that have no equally dense key counterpart. On Quora the two distributions are much more aligned, in line with the dataset's symmetric duplicate-detection setup where queries and keys are both short questions.

**Top-1 MIPS score histograms.** Figure 30 reports the distribution of top-1 MIPS scores $\langle \mathbf{q}, \mathbf{k}^\star \rangle$ on each dataset. Quora is sharply skewed toward 1.0 (mean 0.86, median 0.88), reflecting near-duplicate matches; NQ and HotpotQA are centered well below saturation (means 0.71 and 0.74), with broad Gaussian-like spread. These two views are consistent: the larger the gap between $p_\mathcal{X}$ and $p_\mathcal{Y}$ in Figure 29, the lower the typical $\langle \mathbf{q}, \mathbf{k}^\star \rangle$ in Figure 30, and the more headroom the learned mapping has to improve approximate retrieval, which matches the pattern observed in Section 4.4 and Appendix A.8, where KEYNET delivers its largest gains on NQ and HotpotQA and saturates earliest on Quora.

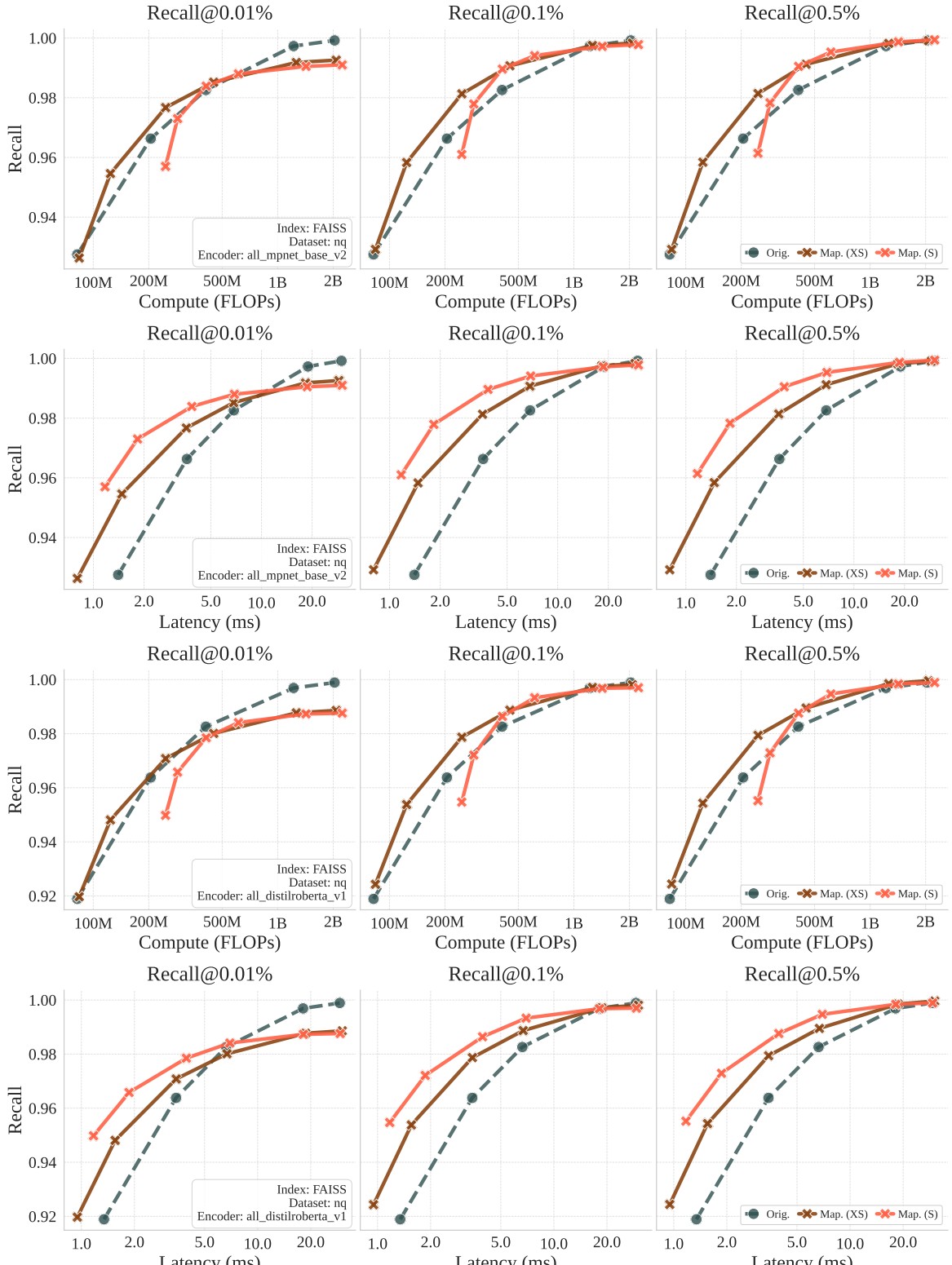

*Figure 12.* FAISS-IVF integration with KEYNET on **NQ** for two $d = 768$ encoders. From top to bottom: MPNet vs. FLOPs, MPNet vs. Latency, DistilRoBERTa vs. FLOPs, DistilRoBERTa vs. Latency. Each row reports Recall@$\{0.01\%, 0.1\%, 0.5\%\}$. Curves compare the original queries (*Orig.*, baseline) against queries mapped by KEYNET at XS and S sizes.

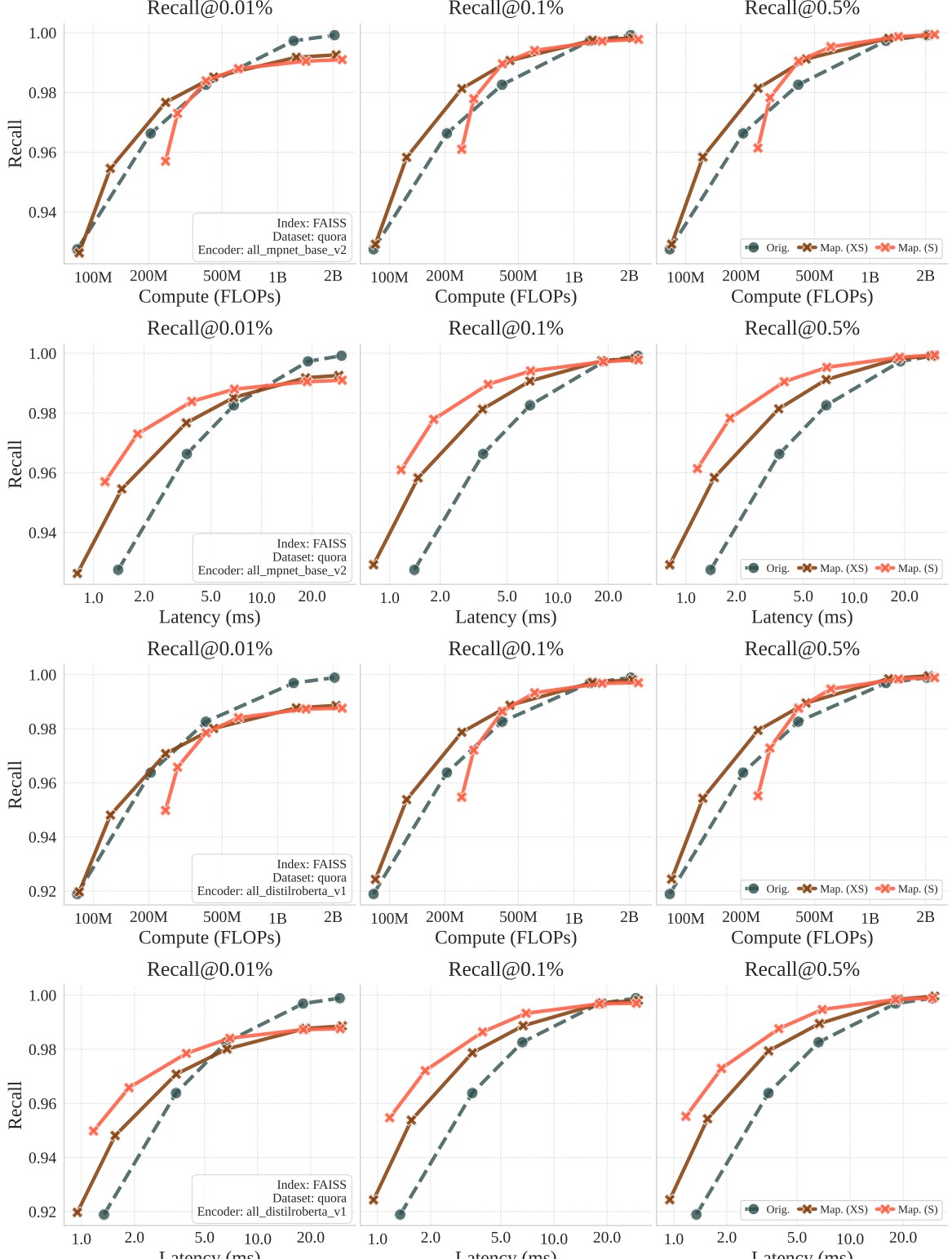

*Figure 13.* Same setting as Figure 12, on **Quora**.

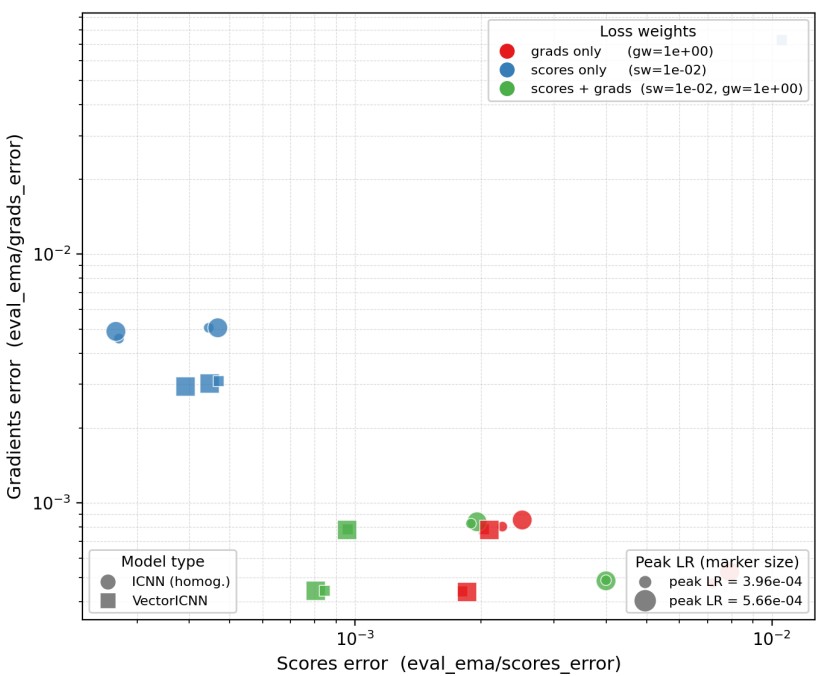

*Figure 14.* Effect of loss weights on the gradient/score error trade-off, on **NQ** with MiniLM at $L = 16$. Colors mark the loss configuration: grads only ($\lambda_{\mathrm{grad}} = 1$, red), scores only ($\lambda_{\mathrm{score}} = 10^{-2}$, blue), and scores+grads (the default, $\lambda_{\mathrm{score}} = 10^{-2}$, $\lambda_{\mathrm{grad}} = 1$, green). Marker shape distinguishes SUPPORTNET (homogenized ICNN, circles) from KEYNET (VectorICNN, squares); marker size encodes peak learning rate.

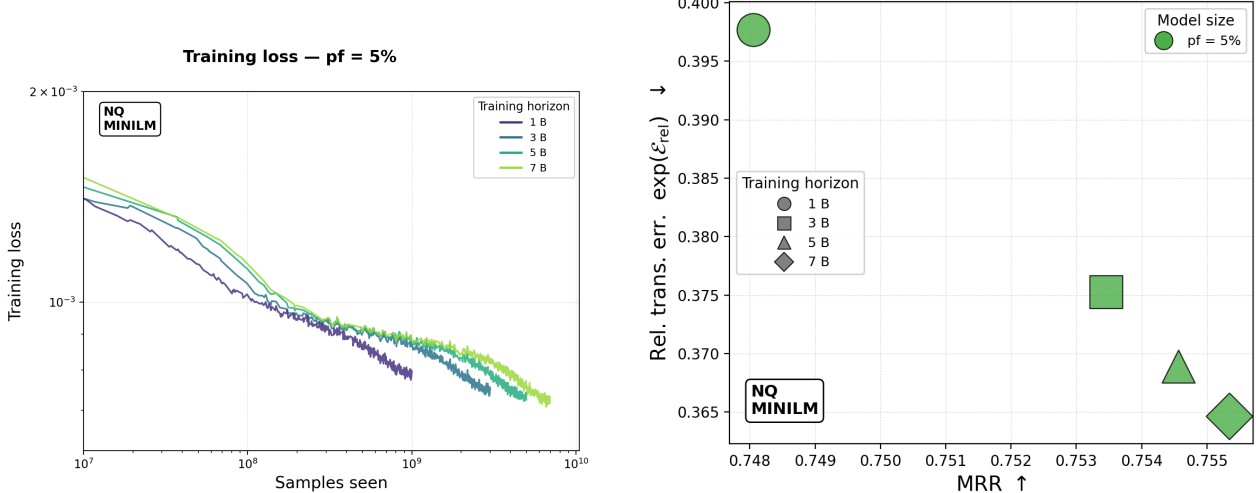

*Figure 15.* Effect of training horizon for a pf=5% (S) KEYNET with $L = 16$ on **NQ**/MiniLM. *Left*: training loss vs. samples seen, for horizons $\{1, 3, 5, 7\}$B (cosine schedule). *Right*: relative transport error $\exp(\mathcal{E}_{\mathrm{rel}})$ vs. MRR at the end of each horizon (lower-right is best). Training loss continues to decrease with longer horizons, but downstream MRR and $\mathcal{E}_{\mathrm{rel}}$ plateau past $\approx 3$B samples.

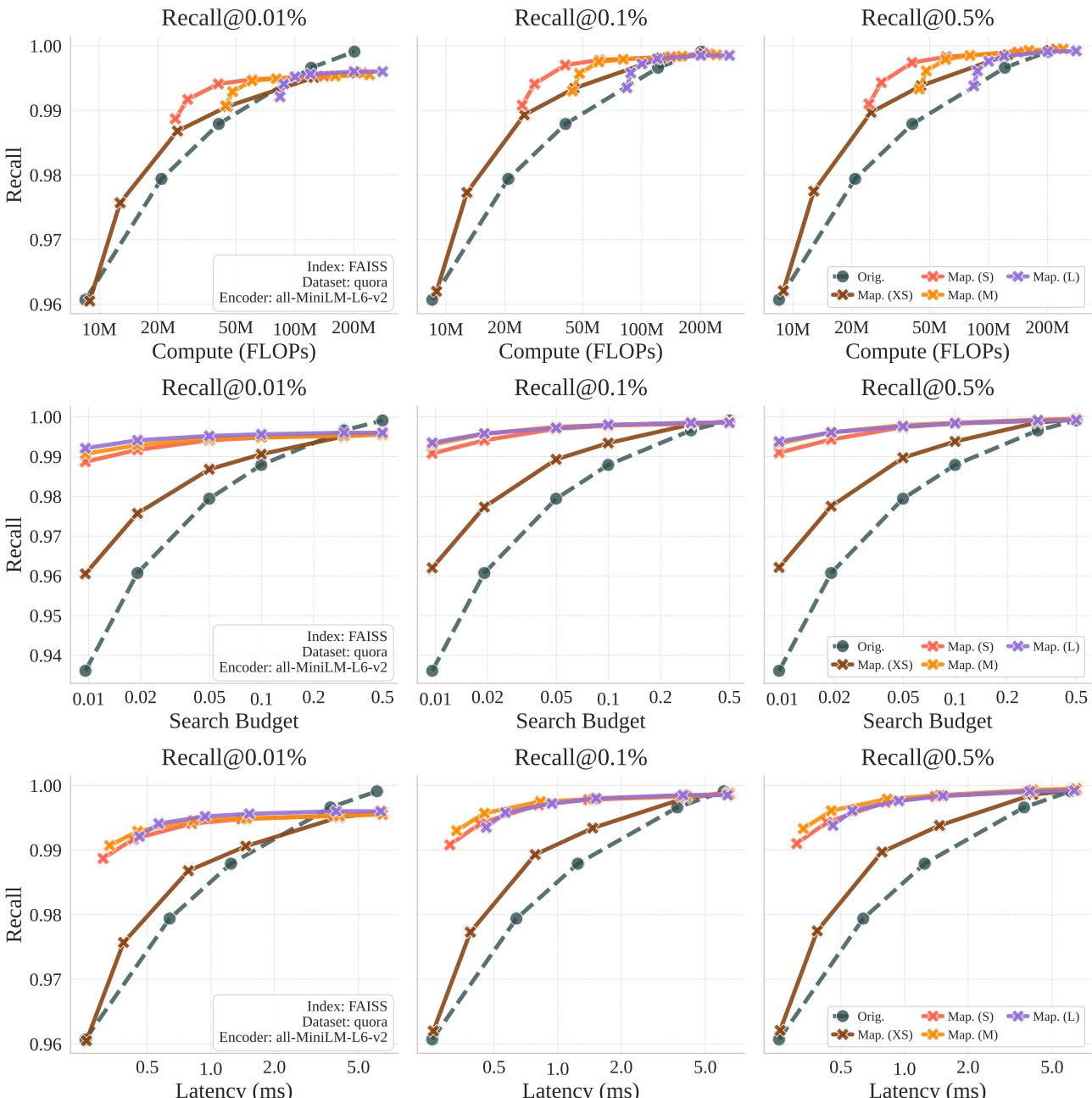

*Figure 16.* FAISS-IVF integration with KEYNET on **Quora** ($n \approx 0.5$M). From top to bottom: cost as compute (FLOPs), search budget (fraction of the database scanned), and wall-clock latency (ms). Each row reports Recall@$\{0.01\%, 0.1\%, 0.5\%\}$ against the corresponding cost axis. Curves compare the original queries (*Orig.*) to queries mapped by KEYNET at XS, S, and M sizes.

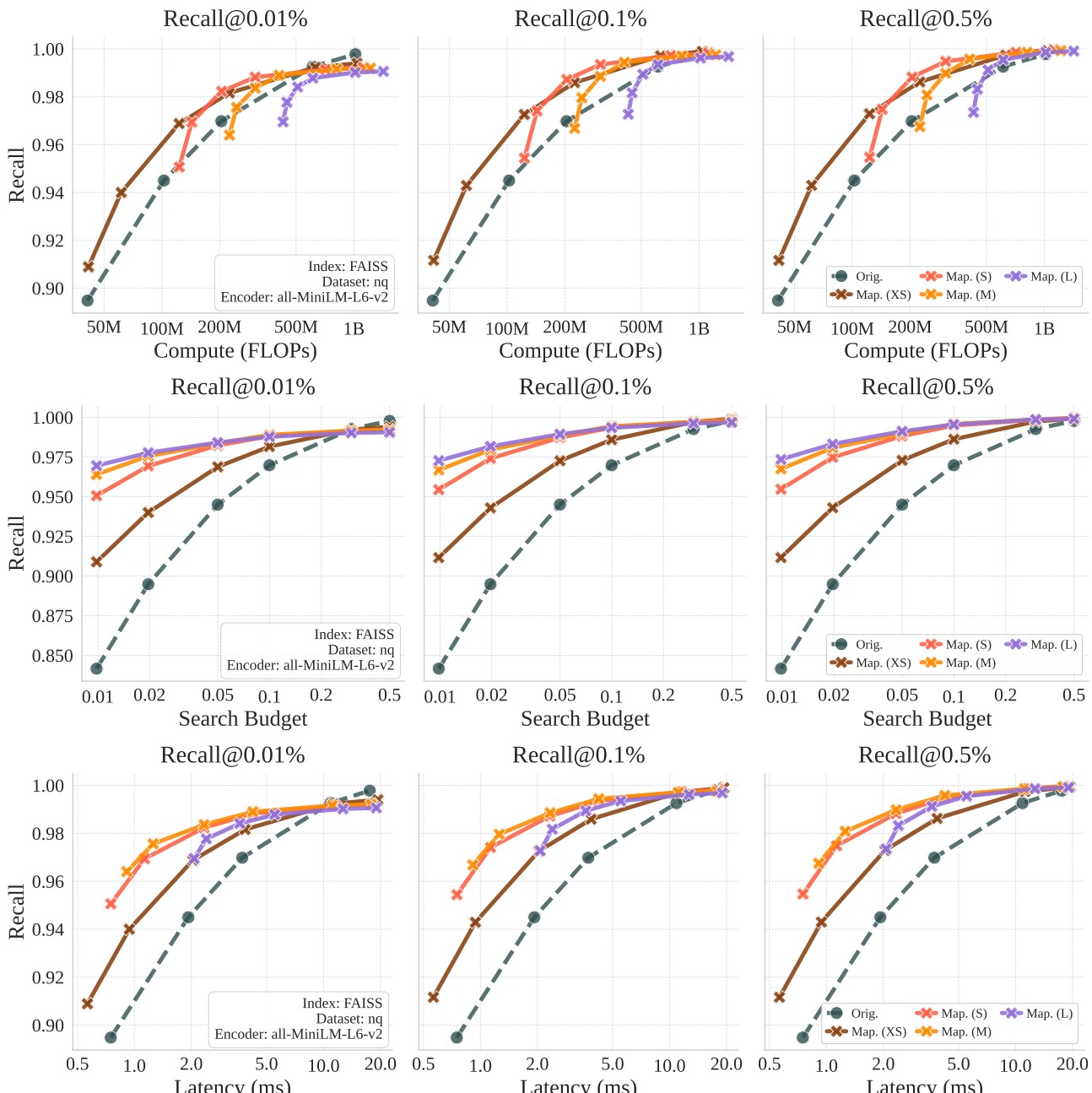

*Figure 17.* Same setting as Figure 16, on **NQ** ($n \approx 2.5$M).

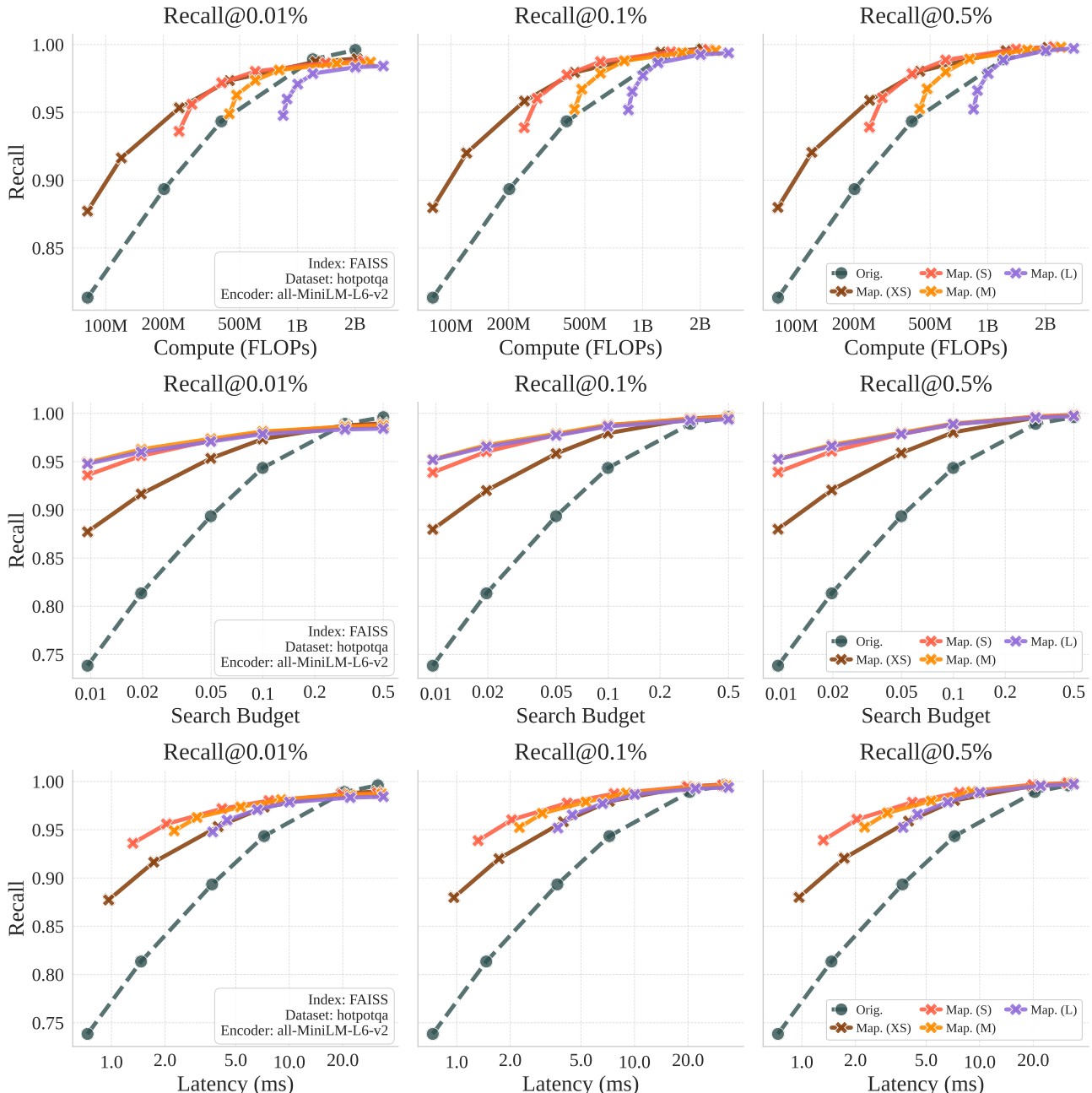

*Figure 18.* Same setting as Figure 16, on **HotpotQA** ($n \approx 5.2$M).

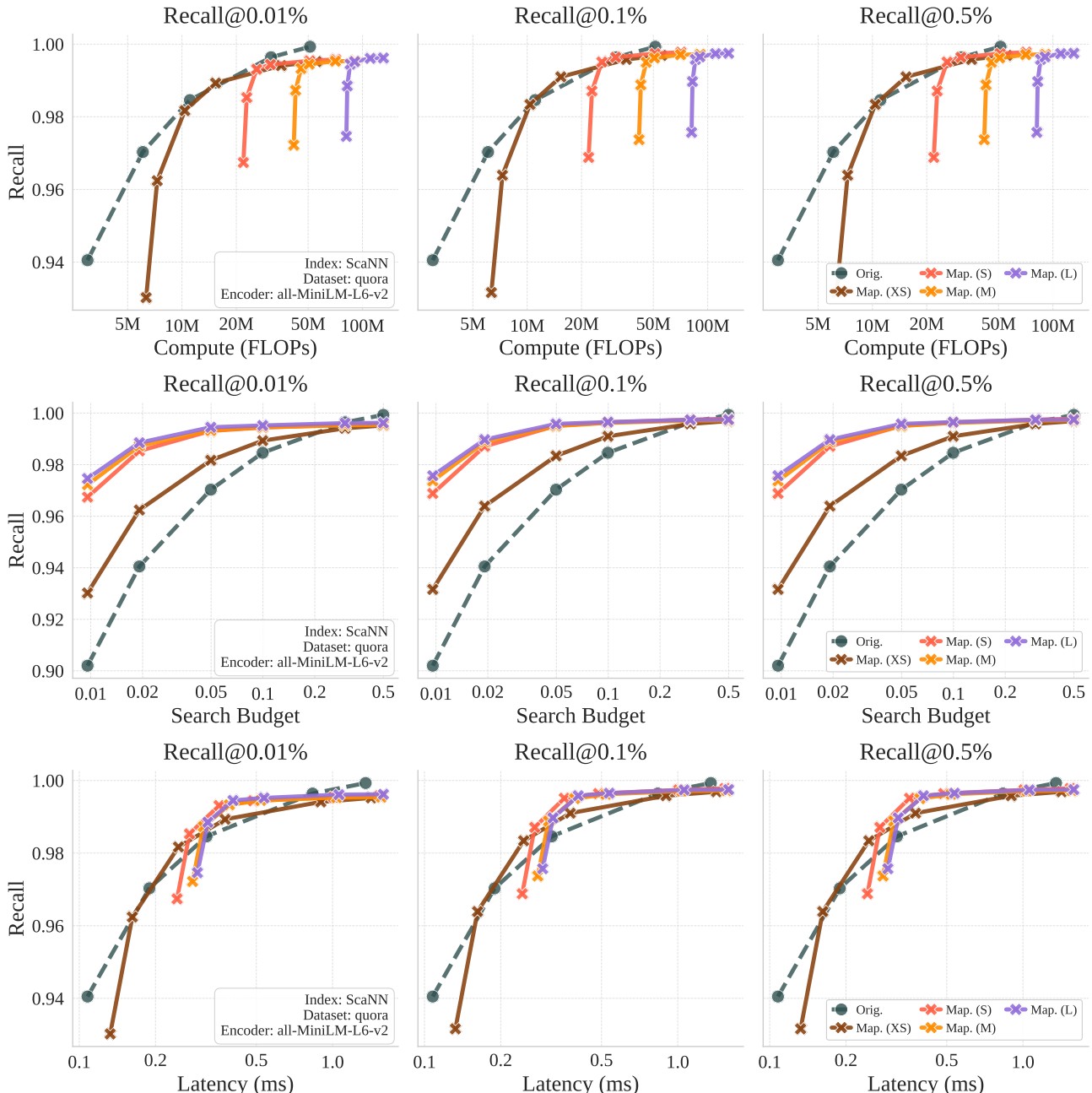

*Figure 19.* ScaNN integration with KEYNET on **Quora**.

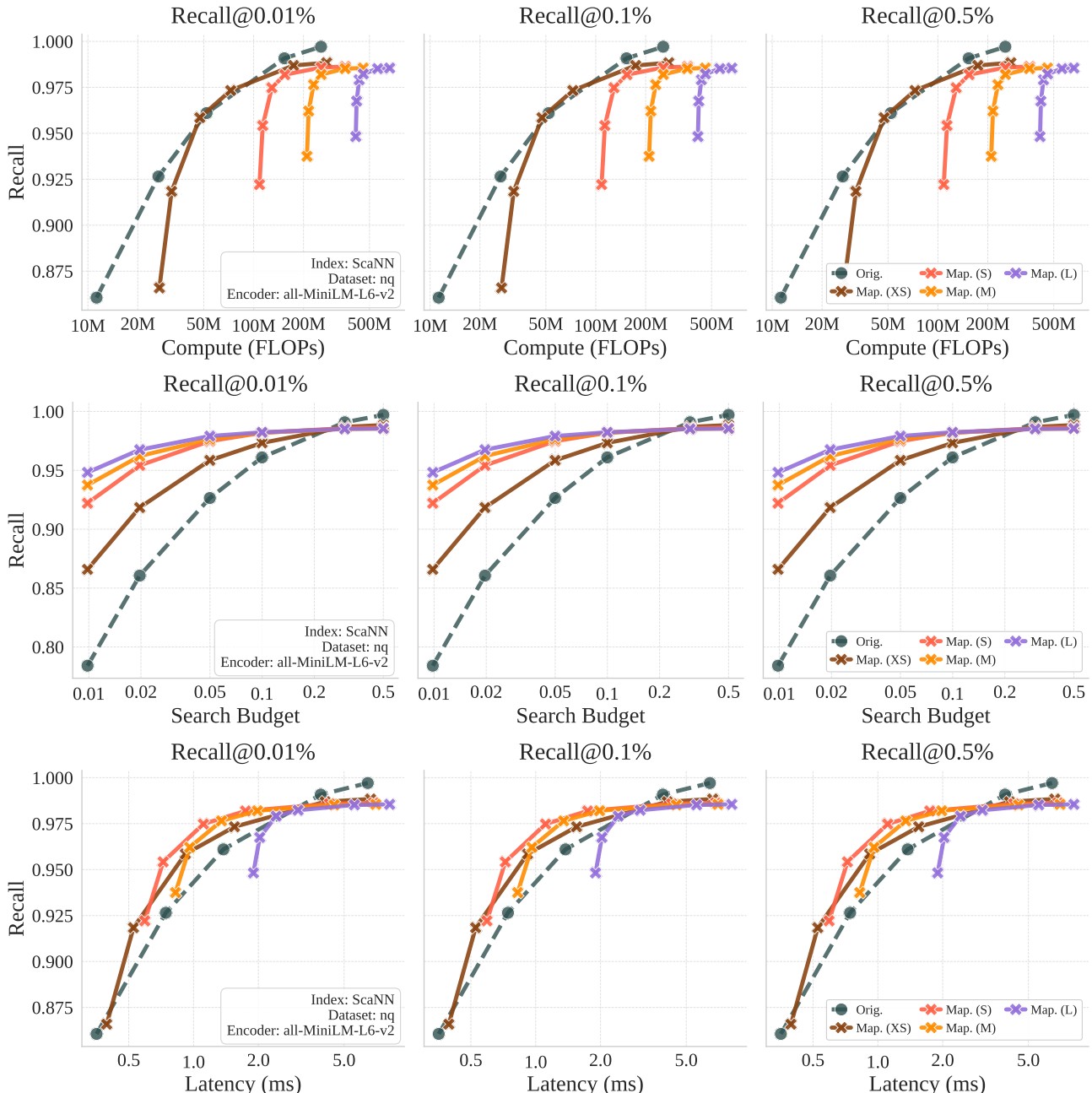

*Figure 20.* ScaNN integration with KEYNET on **NQ**.

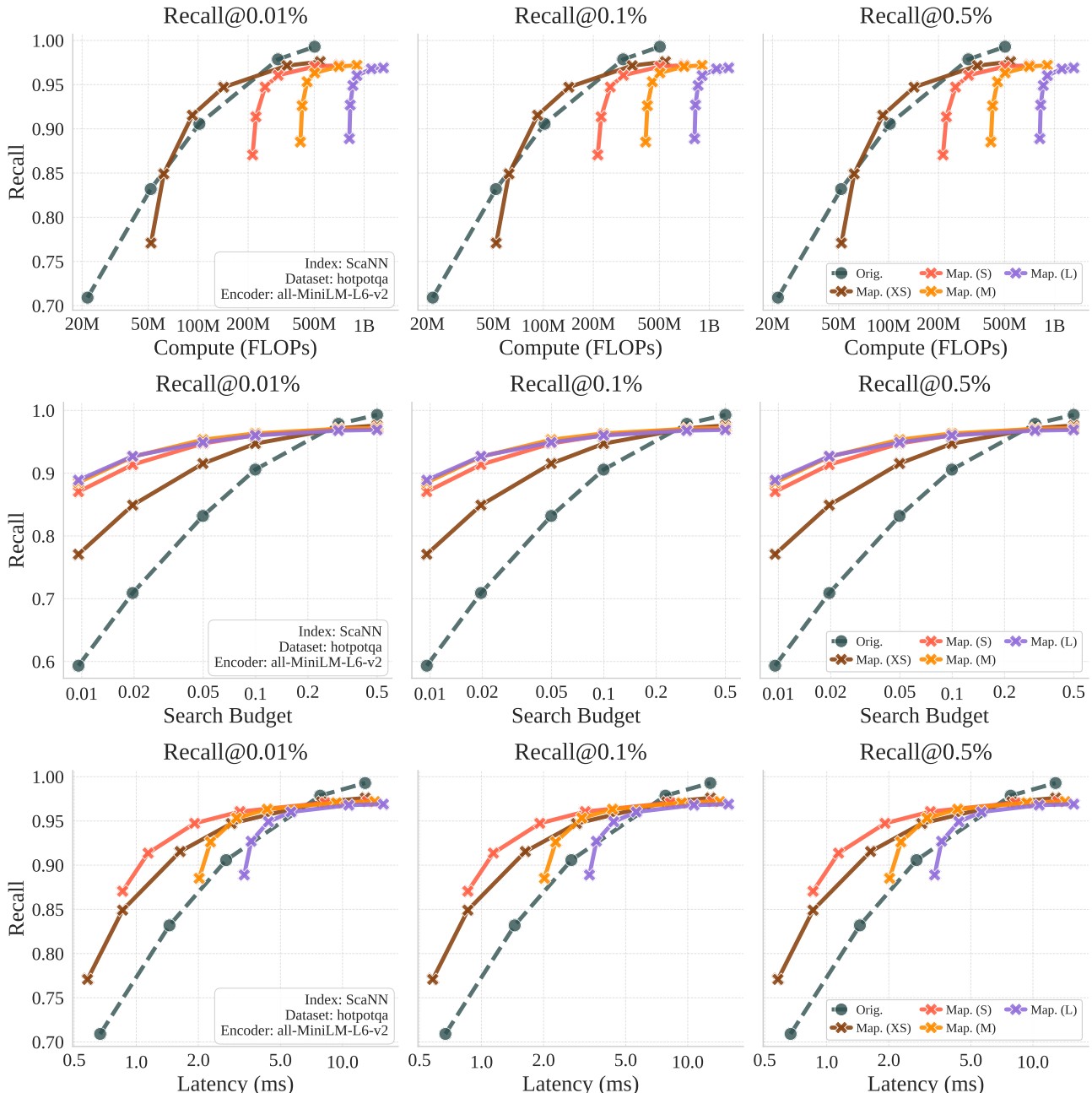

*Figure 21.* ScaNN integration with KEYNET on **HotpotQA**.

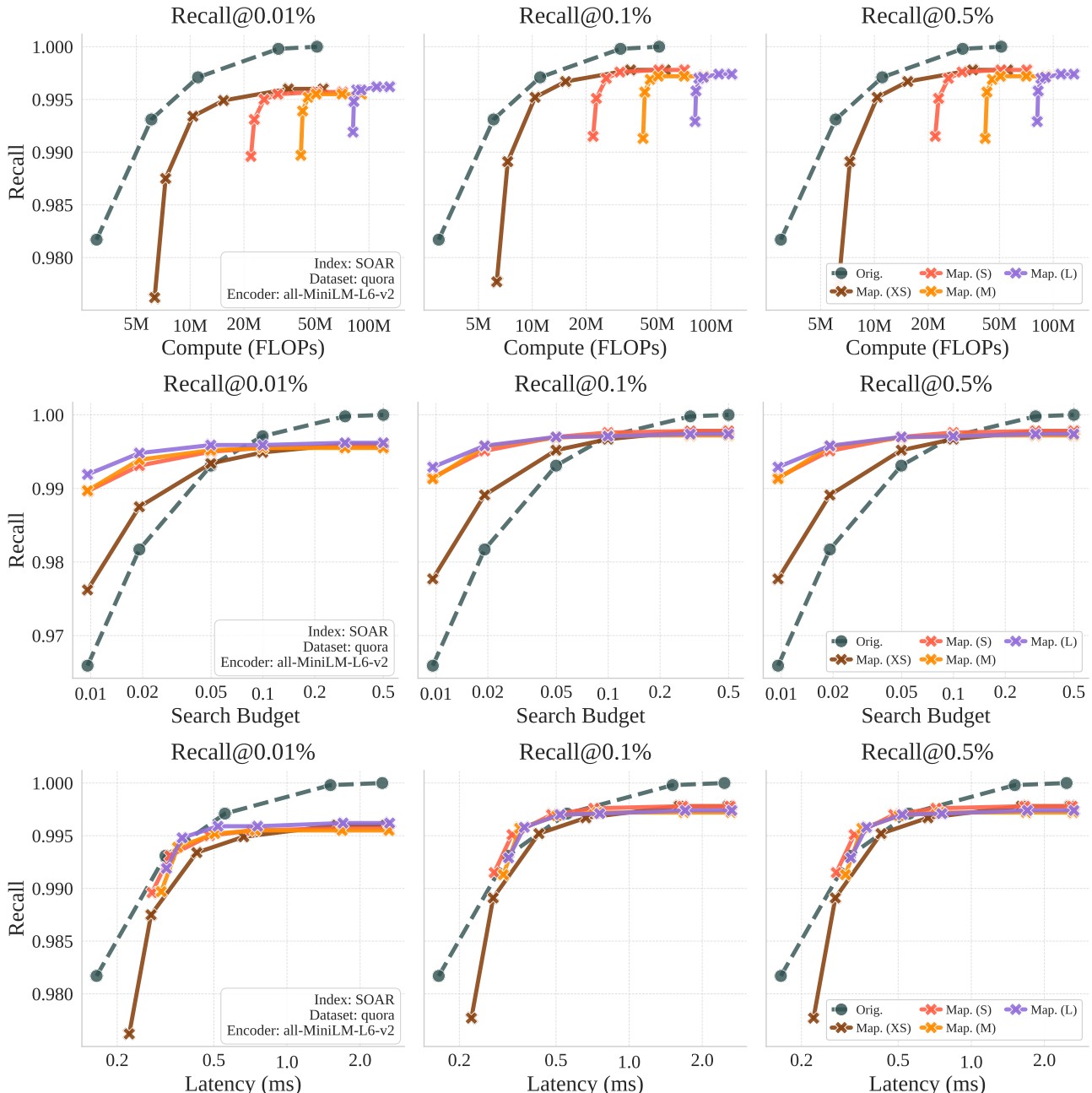

*Figure 22.* SOAR integration with KEYNET on **Quora**.

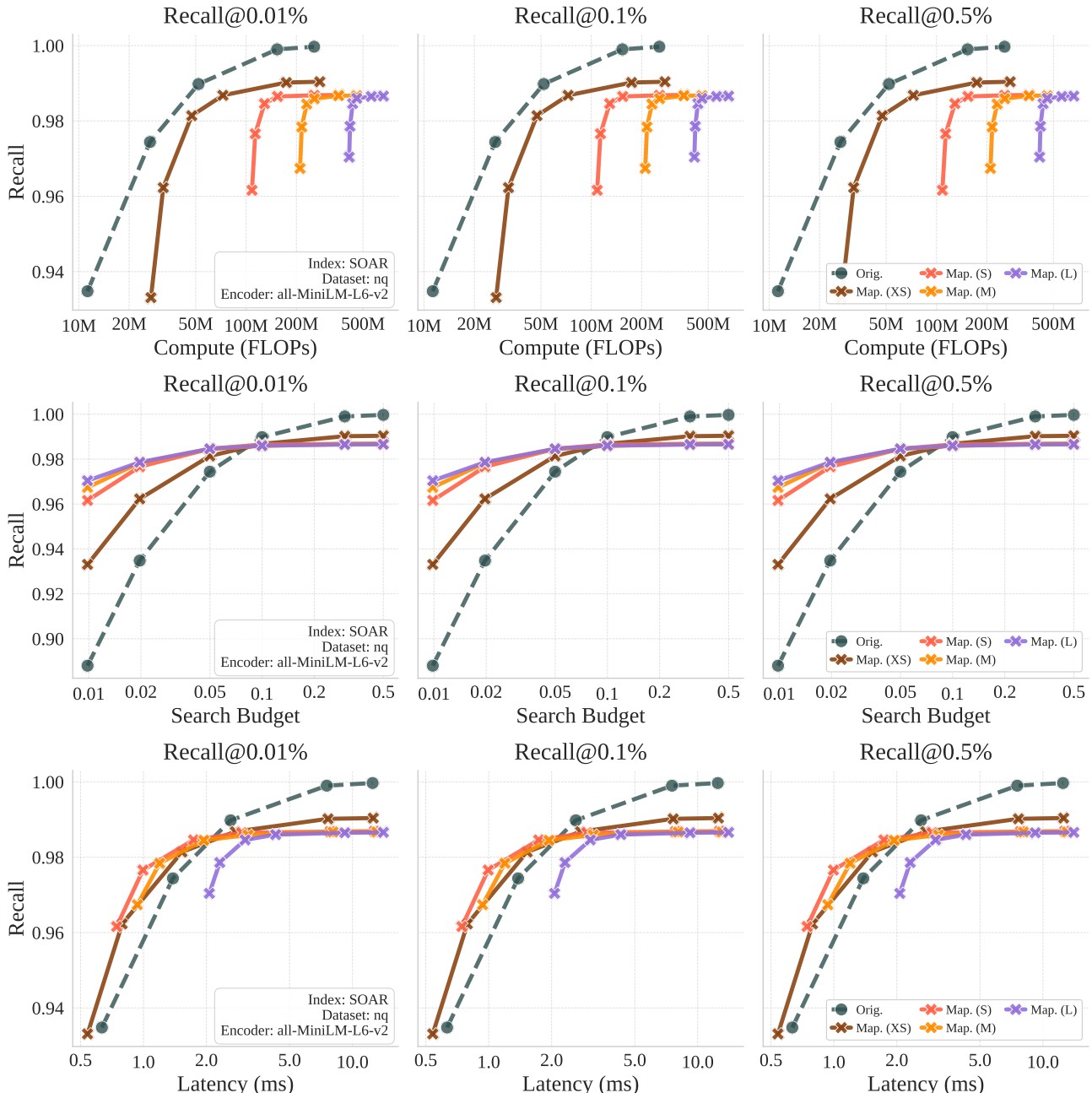

*Figure 23.* SOAR integration with KEYNET on **NQ**.

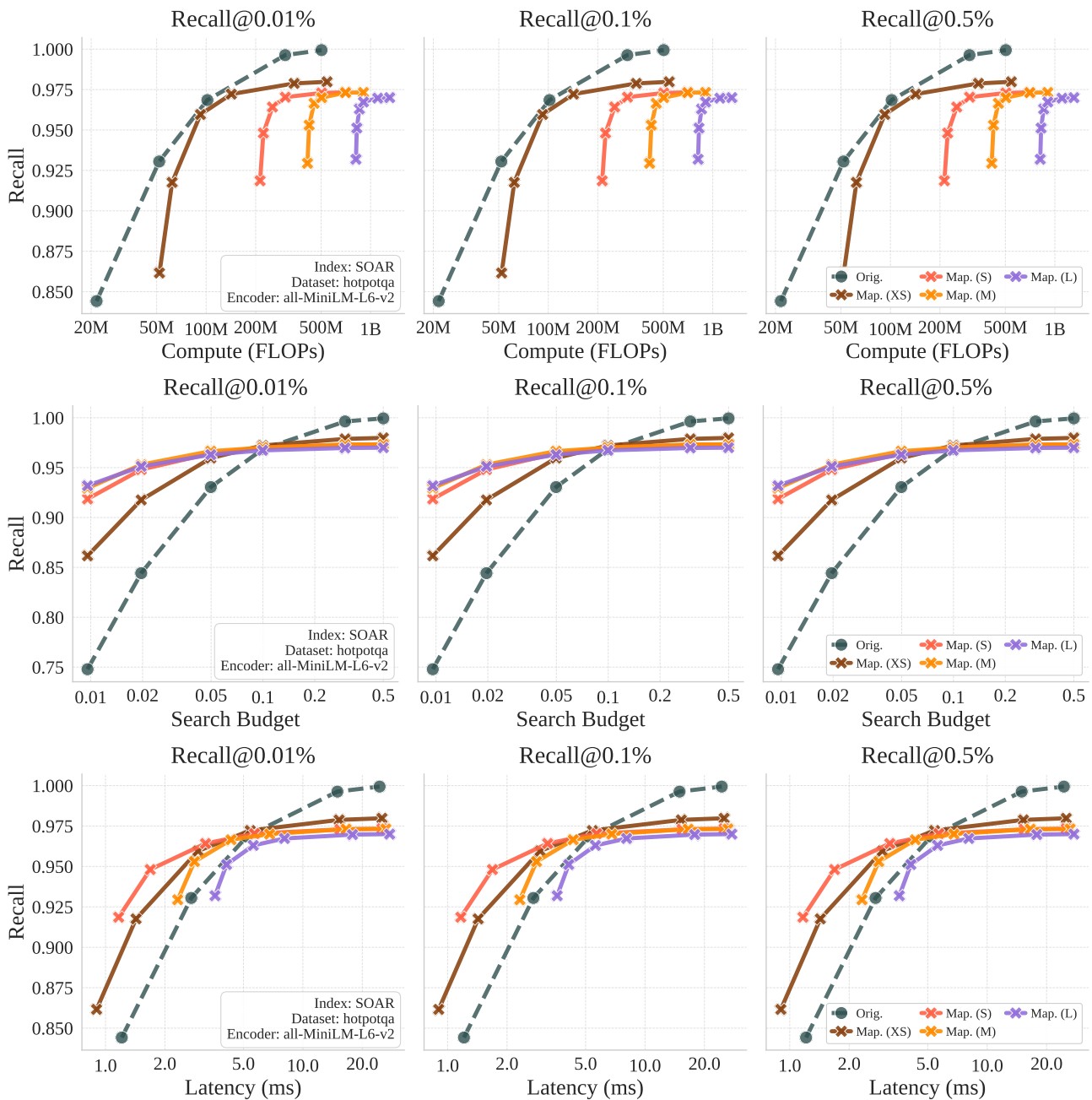

*Figure 24.* SOAR integration with KEYNET on **HotpotQA**.

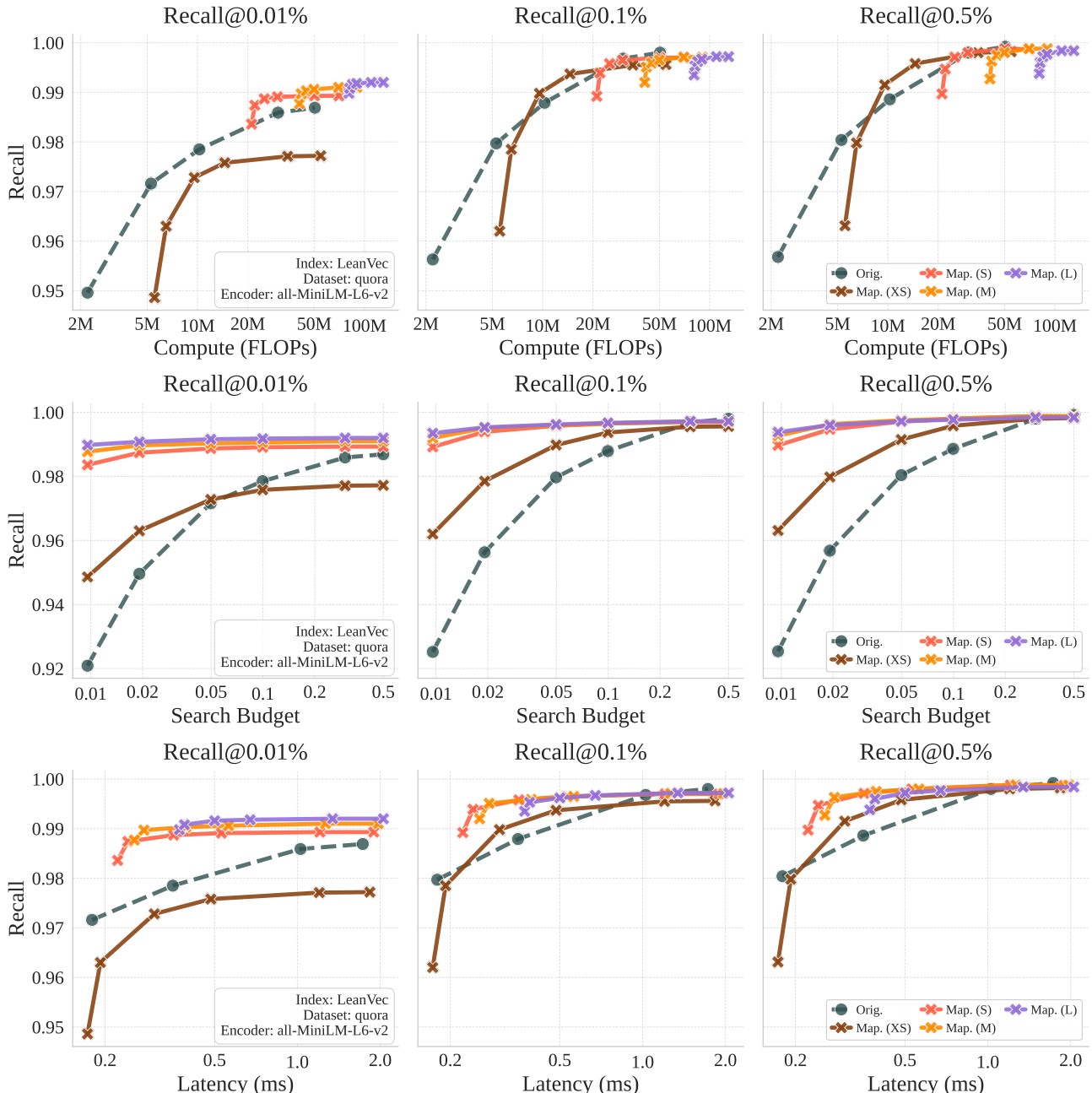

*Figure 25.* LeanVec integration with KEYNET on **Quora**.

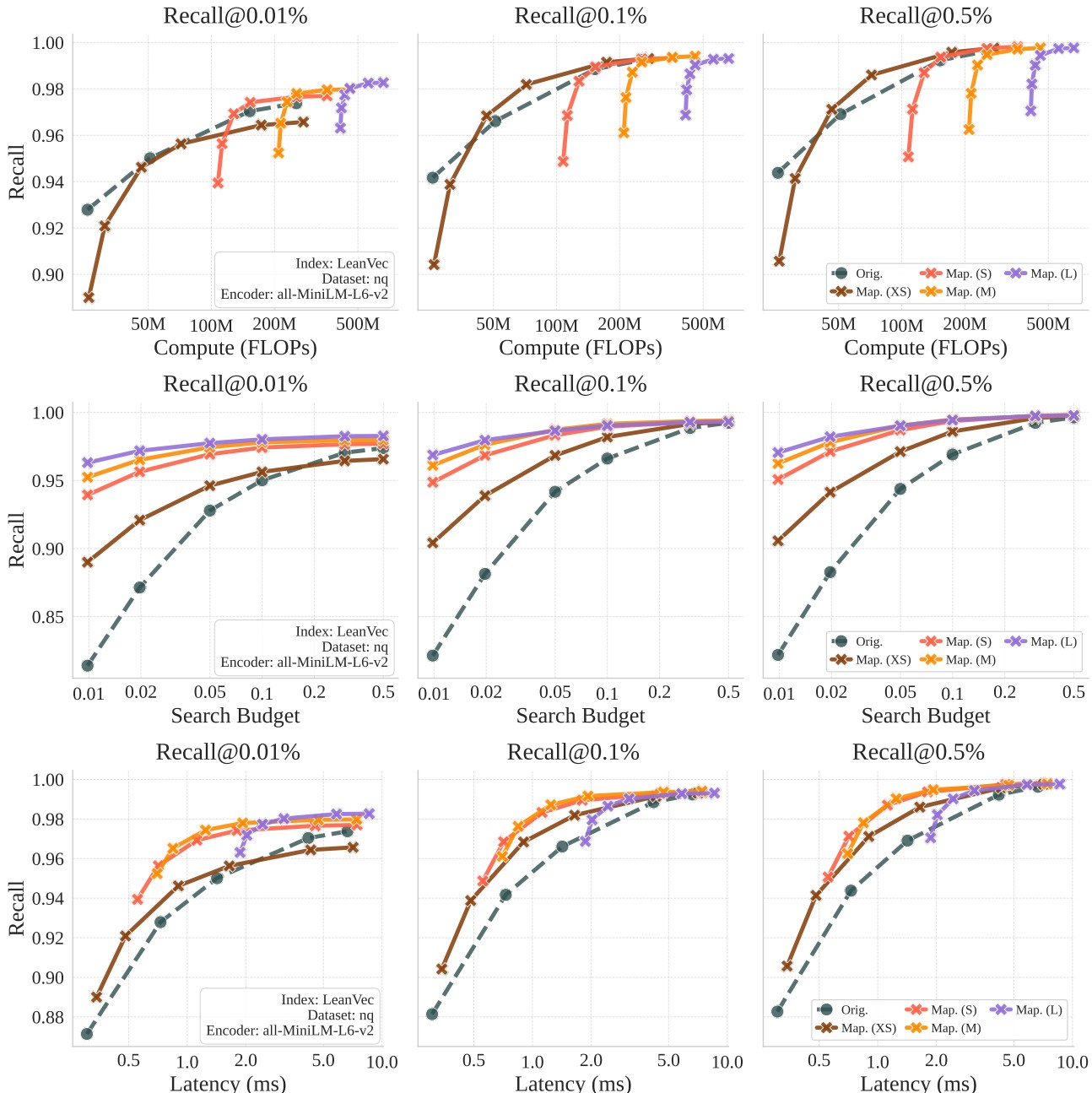

*Figure 26.* LeanVec integration with KEYNET on **NQ**.

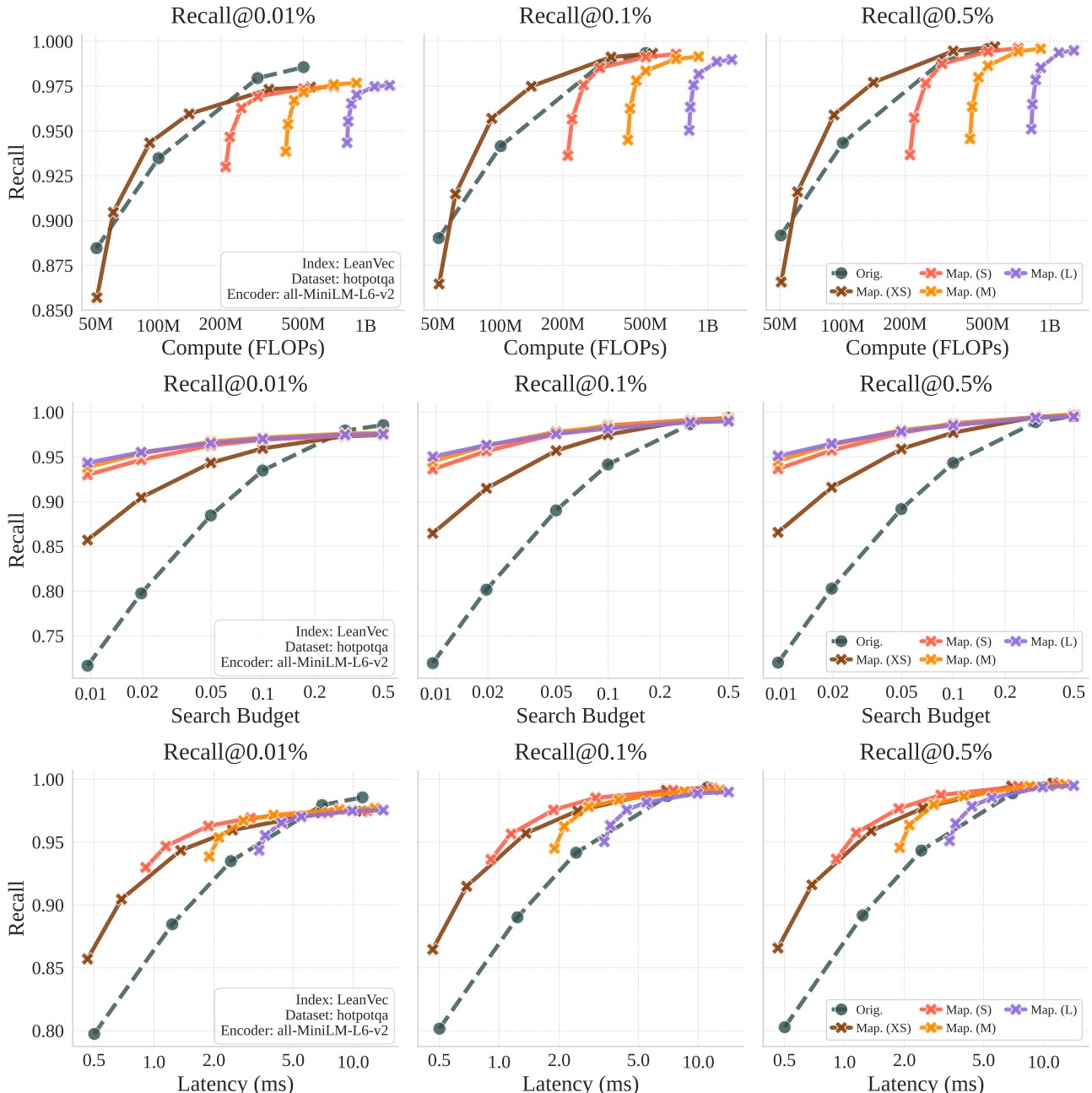

*Figure 27.* LeanVec integration with KEYNET on **HotpotQA**.

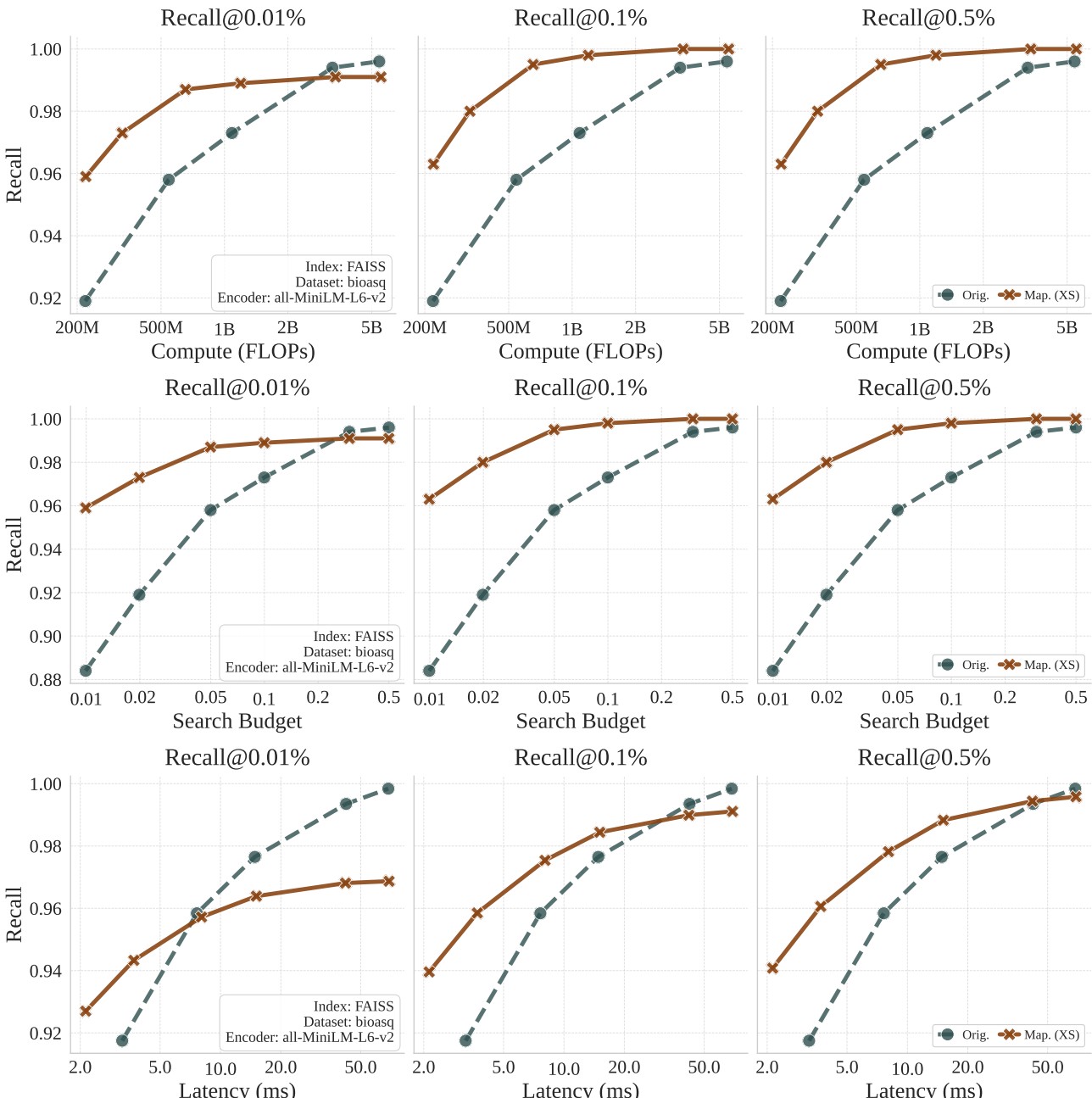

*Figure 28.* FAISS-IVF integration with KEYNET on **BioASQ** ($n \approx 15$M). From top to bottom: cost as compute (FLOPs), search budget (fraction of the database scanned), and wall-clock latency (ms). Each row reports Recall@$\{0.01\%, 0.1\%, 0.5\%\}$. Curves compare the original queries (*Orig.*) to queries mapped by KEYNET at the XS size.

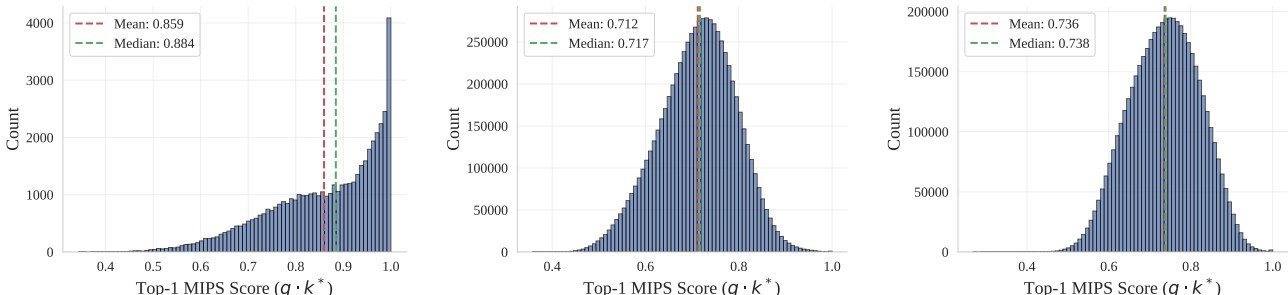

*Figure 29.* Two-dimensional projections of queries and keys for **Quora** (*top*), **NQ** (*middle*), and **HotpotQA** (*bottom*). Each row shows: (a) a joint scatter of both sets, (b) the kernel density estimate of the keys, and (c) the kernel density estimate of the queries. Color-bar magnitudes are not directly comparable across the keys and queries panels (they are individually normalized); compare shape and location instead. All datasets show clear query-side modes with no matching key density.

*Figure 30.* Histograms of the top-1 MIPS score $\langle \mathbf{q}, \mathbf{k}^\star \rangle$ on **Quora** (*left*), **NQ** (*middle*), and **HotpotQA** (*right*). Vertical lines mark the empirical mean and median. Quora concentrates near 1.0 (duplicate detection); NQ and HotpotQA peak around 0.7, indicating substantially larger query/key mismatch and more headroom for an amortized mapping to help.

