# OpenReview forum: "Amortized Maximum Inner Product Search with Learned Support Functions"
_ICML.cc/2026/Conference — ICML 2026 regular_

### Official Review · Reviewer_9vKD · 2026-03-02

**Soundness:** 3
**Presentation:** 3
**Significance:** 2
**Originality:** 3
**Overall Recommendation:** 4
**Confidence:** 3

**Summary:**

The paper focus on Maximum Inner Product Search (MIPS). Considering the query distribution, it proposes a learning-based approach that trains neural networks (NN) to directly predict MIPS results. The key part is that the value function between query and result can be related to the support function from the perspective of convex optimization. Based on this connection, this study design two special NNs to learn value/support function. In the experiment, the method practically requires a existing approximate MIPS indexing for large scale dataset. Though the one time training cost is high, the proposed method does improve the searching efficiency on natural datasets.

**Compliance With Llm Reviewing Policy:**

Affirmed.

**Final Justification:**

The reader might find it interesting to apply NN and convex optimization perspectives to solve MIPS. Practically, this approach does improve the search performance of existing indices like IVF. However, the cost is the introduction of NN training and multiple hyperparameters. While the training cost might be a one-time expense in the pipeline, it is somewhat substantial. Additionally, it introduces more hyperparameters to tune, which increases the time required for the index building pipeline. Moreover, improper hyperparameter settings might even degrade search performance. The rebuttal addresses my main concern about amortization and the question regarding training cost, and I have raised my overall score to 4.

**Key Questions For Authors:**

See W1-W2.

**Limitations:**

yes

**Strengths And Weaknesses:**

**Strengths**

S1: the core idea is interesting that we can use NN and convex optimization perspectives to solve MIPS.

S2: there is a example to help reader quickly understand the main idea.

**Weaknesses**

W1: there is not explicitly complexity analysis, which is significant for MIPS. It is important to analyze query time, indexing time, and index size.

W2: the experiment is not sufficiently convincing. First, it lacks other baseline indexs from traditional MIPS and approximate MIPS methods. Second the evaluation metric is not appropriate and unconventional. It is unusual to use FLOPS as the measure of cost for the MIPS. Third, not only searching time, the time of indexing (training) is also important, which is missing in the experiment.

---

> ### Author Rebuttal · Authors · 2026-03-30
>
> We are grateful for your time reviewing our paper. **There are a few important misunderstandings in your review**. We believe your remarks are however very valuable, as they will help us present better our work to avoid such misunderstandings.
>
> ⭐ Please find **[HERE](http://tiny.cc/qp41101)** a link to new plots⭐
>
> > **W1: there is not explicitly complexity analysis, which is significant for MIPS. It is important to analyze query time, indexing time, and index size.**
>
> Our approach is quite original in the world of MIPS, which might explain your misunderstanding.
>
> We **do not** propose a new indexing method. We take an "amortization" perspective: we **learn** how to predict the top-1 key in one shot, using a NN (See Intro, L.40 to 72).
>
> Equipped with these *amortized* MIPS models, we use their predictions to feed existing indexers (such as FAISS/IVF, or SCANN/SOAR/LeanVEC in our most recent results) with better inputs: we propose to predict **directly** an approximation of the optimal key **(L.61 to 70)** and show how this speeds up MIPS solvers. We will tone down technical aspects in our abstract, and feature more prominently this amortization perspective.
>
> We show results as  query time / performance Pareto fronts in Figure 2 and 4 in the paper (+ appendices).
>
> > **Beside, the paper introduce the NN to MIPS, which makes the additional factor like the NN size worth to be analysis.**
>
> As often in ML, NN sizing is entirely left to the user. This is described in the paragraph **Network sizing** in p.4. We explored many scenarios (various depths `L=8, 16`, total parameters count `XS,S,M,L,XL`, see e.g. **L 293**.
>
> > **First, it lacks baselines from traditional MIPS and approximate MIPS methods, including even brute-force search**
>
> This seems to be an important misunderstanding.
>
> Brute force search is usually reported in the top right corner of our plots (maximal flop count, for a maximal recall).
>
> FAISS indexing is used both when scoring the baseline approach (passing directly $\mathbf{x}$) but also when passing the output of our method ($\hat{\mathbf{y}}(\mathbf{x})$).
>
> Please read again the first paragraph of S.4.5, from **L 418 to 436**. We commit to rewriting this part more clearly, and adding an illustrative plot showing $\mathbf{x}, \hat{\mathbf{y}}(\mathbf{x})$ as inputs to an indexer, this will certainly help forming a better intuition.
>
> ⭐ Please look at **§3** for new eval plots using {SCANN, SOAR, LeanVEC}
>
> > **Second the evaluation metric is not appropriate and unconventional. It is unusual to use FLOPS as the measure of cost for the MIPS.**
>
> We interpret from your comment that you would prefer to see latency or search efforts.
>
> ⭐ Please look at **§3** for new eval plots using alternative compute (time, num probes) metrics.
>
> > **Third, not only searching time, the time of indexing (training) is also important, which is missing in the experiment.**
>
> By bringing an ML perspective to retrieval, we also propose to spend more time learning (train budget) if this yields speed up at inference time. We do observe significant 2x to 4x speed-ups in various regimes and on various datasets. When such MIPS operations are run repeatedly at scale, these inference speed ups justify much larger training budgets than what is typically used for indexing in classic methods.
>
> The queries are augmented 5–100X **(L. 280)** , e.g.:
> - Quora: 0.5M queries × 100 aug. = 50M
> - HotPotQA: 5M queries × 10 aug. = 50M
> We train on 1B–10B samples **(Fig. 7) x-axis**, i.e. 20–200 epochs, on H100 nodes: Takes 6 hours (XS, 1B) to 1 week (XL, 20B).
>
> ⭐ Please look at **§4** and **§7** for corrected Fig. 7 on y-axis.

---

> > ### Author Rebuttal · Reviewer_9vKD · 2026-04-01
> >
> > Thanks for clarifying that the main contribution lies in amortization. The experiment is much clearer and more convincing after more indexing algorithms are added and training cost is reported.

---

> > > ### Author Response · Authors · 2026-04-04
> > >
> > > Dear Reviewer,
> > >
> > > ✅ **Update**: we added in S.9 a schematic figure that illustrates the way `SupportNet` and `KeyNet` could be ultimately used. `SupportNet` for routing, `KeyNet` for approximation and input to other IVF approaches. We will improve these tentative figures, but essentially our goal is to make sure readers will understand very quickly the amortized/learning perspective.
> > >
> > > ---
> > >
> > > We are thankful for your time and efforts reviewing our work. We are also grateful for your criticism, this was very useful to help us frame our paper better. We have used your feedback to improve presentation, in particular:
> > >
> > > - reduce technical aspects in the abstract and contextualize more clearly our approach as a neural amortized approach that is complementary to existing indexing works. Incorporating the feedback from reviewer `wt6o` to focus more clearly on "MIPS for embeddings", we would get something like:
> > >
> > > > Maximum inner product search (MIPS) is a crucial subroutine in machine learning, requiring the identification of key vectors that best align with a given query. We propose *amortized MIPS*: a regression-based approach that trains neural networks to directly predict MIPS solutions, amortizing the cost of repeatedly solving MIPS for queries drawn from a *known* distribution over a *fixed* key database. Our key insight is that the MIPS value function is the *support* function of the key set, a well-studied convex object whose gradient yields the optimal key. This motivates two complementary amortized models: a `SupportNet`, an input-convex network that models the support function and a `KeyNet`, a vector-valued network that directly regresses the optimal key. Both are trained with objectives derived from the geometric structure of support functions. `SupportNet` can serve as a cluster router, steering queries toward relevant database partitions, while `KeyNet` can be used as drop-in replacements for the original query and fed directly to optimized indexing pipelines. Our experiments on the BEIR benchmark show that, for document embeddings, learned `SupportNet` and `KeyNet` can significantly improve IVF match rates when accounting for compute effort, measured in FLOPS, number of probes or time.
> > >
> > > - Emphasize more strongly in experiments the differences between FLOPs / num probes / latency (time) compute metrics, and the different tradeoffs they induce.
> > >
> > > - Add a schematic figure illustrating the routing approach and the query modification approach through amortization.
> > >
> > > Please do not hesitate relaying any other question you may have to the AC.
> > >
> > > Many thanks again for your time.
> > >
> > > The Authors

---

### Official Review · Reviewer_Zca3 · 2026-03-09

**Soundness:** 2
**Presentation:** 3
**Significance:** 2
**Originality:** 2
**Overall Recommendation:** 4
**Confidence:** 3

**Summary:**

This paper proposes amortized Maximum Inner Product Search (amortized MIPS), whose core idea is to train neural networks to predict the solution of MIPS for a given query directly and thus the computational cost of search across queries drawn from a known distribution can be amortized. More specifically, two methods are proposed: (1) SUPPORTNET that learns the convex support function (i.e., the value of the maximum inner product) using Input Convex Neural Networks (ICNNs) and recovers the optimal key via computing its gradient; (2) KEYNET that learns to predict the optimal key directly directly. During the training stage, both SUPPORTNET and KEYNET are trained to matching the support function and the optimal key. During the inference stage, the predicted key is generated either by computing the gradient of the output of SUPPORTNET w.r.t. its input or by using the output of KEYNET directly. The predicted key is then used to retrieve the final result via exact search or approximate search like IVF in Faiss. Experiments on benchmark datasets demonstrate the effectiveness of the proposed method.

**Compliance With Llm Reviewing Policy:**

Affirmed.

**Final Justification:**

The authors' rebuttal have addressed most of my concerns and I raise my rating

**Key Questions For Authors:**

1. As mentioned in the Weakness section, the proposed method introduces a learnable query transformation method but still relies on a standard MIPS algorithm. What is the fundamental advantage of learning this transformation versus directly learning a more efficient relevance function (e.g., training a neural network to approximate the score of the optimal key and using that for ranking directly)? Why is the two-step process (query embedding transformation & MIPS) necessary?
2. The definition in Equation (7) seems weird. If the prediction $\hat{y}$ is perfect, the ratio becomes zero, and log⁡(0) is -inf. How is this metric handled in practice?
3. The paper assumes a static database and a fixed query distribution. In many real-world scenarios, items (i.e., keys) are continuously added or removed, and query distributions can drift over time. How would the proposed amortized MIPS approach handle such dynamic environments?

**Limitations:**

yes

**Strengths And Weaknesses:**

Strengths:
1. MIPS is a core challenge in real-world applications like information retrieval and recommender systems. Leveraging a known query distribution to amortize the search cost is an interesting and valuable idea.
2. The paper is well written and easy to follow. The connection between the MIPS value function and its gradient is explained clearly and the idea of using ICNNs is well motivated. The derivation of SUPPORTNET and KEYNET is easy to understand.

Weaknesses:
1. The proposed amortized MIPS method does not change the fundamental nature of MIPS, which does not support the main claim of amortizing the MIPS process well. It still relies on a standard MIPS algorithm (e.g., exact search or IVF in Faiss) to find the actual item/document (i.e., key) in the database while replacing the original query embedding $x$ with a new vector $\hat{y}$ which is the output of SUPPORTNET or KEYNET. The contribution is therefore better framed as a learnable query transformation method since the core MIPS algorithm keeps unchanged, which deteriorates the claimed novelty significantly.
2. The primary baseline for Figure 4 is a standard FAISS IVF index using the *original* query, which is not strong enough to validate the method's practical benefits. To truly understand the value of the learned transformation, authors should compare against MIPS methods that also leverage query distributions (e.g. ScaNN).
3. Using Flops as the only cost metric in Figure 4 is problematic. It combines the highly optimized and often hardware-accelerated operations of a library like Faiss with the more general-purpose operations of a neural network forward pass. This makes the comparison unfair and not representative of real-world latency. Reporting empirical wall-clock time on a standard CPU/GPU setup is necessary to measure practical speedup.
4. The claim that previous methods "do not leverage information about the query distribution—queries are treated as arbitrary vectors" is inaccurate. Modern approximate MIPS methods like ScaNN do leverage information about the query distribution for index construction through anisotropic quantization loss.

---

> ### Author Rebuttal · Authors · 2026-03-30
>
> We thank you for the constructive review. We have added baselines following your comments.
>
> ⭐ Please find **[HERE](http://tiny.cc/qp41101)** a link to new plots⭐
>
> > **The proposed amortized MIPS method does not change the fundamental nature of MIPS…contribution is… better framed as a learnable query transformation method**
>
> Thanks for this great question. We tip-toed around this in our paper and did not express our views clearly. We believe we have now a better story, thanks to your comment and also based on ongoing work.
>
> Quoting from a recent textbook's abstract (https://arxiv.org/abs/2202.00665), *Amortized optimization methods use learning to predict the solutions to [optimization] problems*.
>
> For MIPS against a $n\times d$ dataset, amortization can mean either
> - predicting an index (integer $\leq n$), or
> - the top key (a $d$ vector),
> - the top score (a real).
>
>  We chose to amortize top key and score.
>
> From there, the user can use score for routing, or key $\hat{\mathbf{y}}(\mathbf{x})$ to either:
> - (1) identify a "real" key *in* the dataset closest to $\hat{\mathbf{y}}(\mathbf{x})$, i.e. run a MIPS optimizer with a better initialization.
> - (2) use $\hat{\mathbf{y}}(\mathbf{x})$ "as-is" in a downstream task, as a "virtual" point (e.g. to define a loss or a score). Here optimization is replaced entirely by an amortized function, even if their outputs do not match strictly.
>
> We focus on (1) in this draft as more immediately impactful; (2) is ongoing but more speculative.
>
> > **…authors should compare against MIPS methods that also leverage query distributions (e.g. ScaNN).**
>
> ⭐ Please look at **§3** for new eval plots using {SCANN, SOAR, LeanVEC}
>
> > **Reporting empirical wall-clock time on a standard CPU/GPU setup is necessary to measure practical speedup.**
>
> ⭐ Please look at **§3** for new eval plots also reporting latency (ms) and #probes.
>
> > **The claim that previous methods "do not leverage information about the query distribution—queries are treated as arbitrary vectors" is inaccurate.**
>
> We agree. We apologize for our sloppy writing in **L.33 Col.2, L.94 col 2**, we rewrote all of this, adding references suggested by `Rev. wt6o`
>
> We meant to use the query distribution to motivate a learning-based paradigm. Existing approaches take a "macro" perspective on it (e.g. linear transforms, better quantization), not the "micro" approach of models with millions of parameters.
>
> > **What is the fundamental advantage of learning this transformation versus directly learning a more efficient relevance function (e.g., training a neural network to approximate the score of the optimal key and using that for ranking directly)?**
>
> Here is our best interpretation of your suggestion.
>
> Given a database of $n$ vectors in dim $d$, our approach predicts either the score of the optimal key (what you suggest above). A dual approach (akin to relaxing the set of $n!$ permutations into the ranking polytope of size $n$) would be to learn a map from input $\mathbf{x}$ into the ranking polytope.
>
> While interesting, we are not sure how to make this work efficiently with NNs. The output dimension would be $n$, not $d$, with the issue that $n\gg d$ in general. If $n$ were to change (new keys), the NN would have to be completely retrained.
>
> > **Why is the two-step process (query embedding transformation & MIPS) necessary?**
>
> As mentioned above, amortization yields a first good guess. If one *needs* an integer index (i.e. a point in the database), one must recover it. The goal of amortization is to show that such a recovery happens faster when using a better starting point.
>
> If the top-1 key can be used directly (e.g. in a reconstruction loss), the prediction of our models could be used "as-is". In many of our experiments, $\mathcal{E}_\text{rel}$ (Eq. 7) can go as low as -2: The geometric average of the reconstruction ratio is exp(-2) approx. 13%, so reconstruction of top-1 key is highly accurate.
>
> > **Equation (7) seems weird. If the prediction is perfect, the ratio becomes zero, and log⁡(0) is -inf. How is this metric handled in practice?**
>
> $\mathcal{E}_\text{rel}$ is an evaluation metric, **not a loss**.
>
> Please read L. 285 (right column). While we track $\mathcal{E}_\text{rel}$ in our experiments, it might be easier to think in terms of its *exponential* (the geometric average of ratio), which is 0 when the metric is -inf.
>
> ⭐ Please look at **§7**: due to a mistake when exporting from wandb, the y-axis values in Fig. 7 ($\mathcal{E}_\text{rel}$) are off by a factor of 2. We apologize.
>
> > **The paper assumes a static database and a fixed query distribution…How would the proposed amortized MIPS approach handle such dynamic environments?**
>
> ⭐ Please look at **§5**: we study robustness to query distribution shift.
>
> More generally, while we did not elaborate on these aspects, they are one of the strongest arguments in *favor* of framing MIPS as a pure learning problem. that could be handled using e.g. continual training.

---

> > ### Author Rebuttal · Reviewer_Zca3 · 2026-04-04
> >
> > The authors' rebuttal have addressed most of my concerns and I raise my rating

---

> > > ### Author Response · Authors · 2026-04-04
> > >
> > > Dear Reviewer,
> > >
> > > ✅ **Update**: In response to your very first question, as well as similar comments from Reviewer `9vKD` we haved added in S.9 of the supplementary material a schematic figure that illustrates the way `SupportNet` and `KeyNet` can be used: `SupportNet` for routing, `KeyNet` for approximation and input to other IVF approaches. These are just tentative figures and must be much improved, but they capture what we would like our readers to take away quickly from the amortized/learning perspective.
> > >
> > > ---
> > >
> > > We are happy to hear that our rebuttal addressed your concerns.
> > >
> > > We are thankful for your critiques and pointy questions. They helped us get a better sense of what aspects of our approach should be more emphasized in the abstract, introduction, related works and conclusion.
> > >
> > > Many thanks for your requests for additional experiments, we are happy to hear that you found them convincing. While we may not have other opportunities to interact during this rebuttal, please feel free to relay other questions to the AC should you have any before the end of the discussion period.
> > >
> > > The Authors

---

### Official Review · Reviewer_wt6o · 2026-03-12

**Soundness:** 2
**Presentation:** 2
**Significance:** 2
**Originality:** 3
**Overall Recommendation:** 2
**Confidence:** 5

**Summary:**

The authors propose training an input convex neural network to predict solutions to the maximum inner product search problem. Specifically, they propose two approaches: (1) SupportNet, that directly predicts the MIPS value of the query and the corresponding key is obtained by gradient at the query; (2) KeyNet, that directly predicts the optimal MIPS key.

In their experiments, the authors demonstrate two potential advantages that can be gained using the proposed approach: improved recall-FLOPs tradeoff compared to an IVF index when used as a query routing mechanism and when searching an IVF index using the predicted optimal key.

**Compliance With Llm Reviewing Policy:**

Affirmed.

**Final Justification:**

The rebuttal addressed some of my concerns regarding the role of SupportNet and the discussion of related work. However, I still remain concerned about the experiments and the viability of the method. I have carefully read the authors' reply but there seems to still be a misunderstanding as I was not talking about the cost of comparing a query to the centroids, but rather the total number inner product evaluations for a given query. The query cost should grow sublinearly with respect to the dataset size (typically the query cost of graph methods grows logarithmically) but the network size, i.e. inference cost, in the paper grows linearly. This cost is hidden in the experiments by the moderate-sized datasets combined with using far less clusters than optimal.

A further issue with the article is with the positioning of the work as solving MIPS but considering only normalized vectors. In their reply, the authors have indicated that they could change the framing as specifically MIPS for embeddings, but I think this would be both a substantial revision and also quite weird as normalized vectors is the one case where a MIPS method is not needed (because in that case the problem is equivalent to regular ANN search with Euclidean distance). Overall, the current version of the paper does not make a convincing case that the method is practical or would lead to practical methods in the future, given the added complexity, the expensive training (indexing) cost, and the above query time limitation. I would encourage the authors to systematically pursue the OOD aspect where the corpus and query distributions differ significantly to potentially demonstrate significant advantages from actually fitting to the query distribution.

**Key Questions For Authors:**

1. How much training data is used in your experiments and long does training the networks take and on which hardware?

2. How much effect do the different terms in the loss functions actually have, i.e. are both terms in the loss functions necessary?

3. Can you clarify the setup of the experiment in Section 4.5, in particular whether the number of clusters was also set to 10 in this experiment and do you query the actual IVF index using the predicted vector (with no reranking based on the original query) or just choose the clusters based on the predicted vector?

**Limitations:**

Limitations are discussed in the paper but not comprehensively; see above. Broader impact is discussed appropriately.

**Strengths And Weaknesses:**

The paper proposes an intriguing new approach to a foundational problem that is increasingly important in modern AI applications and the paper is well-written.

However, the paper does not provide evidence that the chosen approach would have practical value or would provide insights for future research that will have such impact. While SupportNet is conceptually neat, it is unclear why it should be used as it seems inferior in practice to KeyNet. While the authors' experiments demonstrate a small improvement from KeyNet over plain IVF, speeding up IVF using some approximation is of course not new or surprising. It is not convincing that a more expensive and complicated neural network training strategy for predicting the optimal key would be preferable to simpler approximation strategies such as directly predicting the clusters to search [1] or speeding up the within-cluster search by approximating (predicting) the inner products directly using e.g. product quantization or low-rank methods [4, 5] (note that the latter ones also adapt to the query distribution). Further, since the proposed networks predict the support function or the optimal key, it is not clear how they would extend to $k$-MIPS which is of course more relevant in practice.

The experimental results are also not convincing. The experiments set the number of clusters as c = 10 which is much lower than makes sense, in practice this value should be $O(\sqrt{n})$ (e.g. [2]) so I would expect it to be in the hundreds for the smaller datasets and in the thousands for the bigger datasets. It is not clear that training scales to large values of $c$ and the smallest model the authors consider is 1% of the dataset size which is already more than the amount of points you would expect to scan for the considered datasets using reasonable IVF hyperparameters.

A further problem with the experiments is that they use only one embedding model for all datasets. When presenting a learning-based method, I would expect to see generalization and robustness to different models be demonstrated. Also, the used embedding model produces normalized vectors so it is not representative of MIPS use cases where the magnitudes can differ drastically.

It is also not evident that the query and database distributions actually differ just because the queries are shorter texts. You should quantify this difference and ideally use a dataset where the difference is likely to be larger, e.g. by using an instruction-tuned embedding model that actually embeds queries and passages differently or by considering cross-modal retrieval where the queries and database items are from different modalities (e.g. the Yandex text-to-image dataset from Big ANN benchmarks [3]). This would demontrate that there is actually benefit from adapting to the query distribution.

Finally, the work is not contextualized properly. While the proposed specific approach is novel, there exists many prior works that can also adapt to the query distribution [4-8] and these works are not discussed in the paper.

[1] Vecchiato et al. A Learning-to-Rank Formulation of Clustering-Based Approximate Nearest Neighbor Search. SIGIR (2024).

[2] Douze et al. The Faiss Library. IEEE Transactions on Big Data (2025).

[3] Simhadri et al. Results of the Big ANN: NeurIPS'23 competition. NeurIPS (2025).

[4] Tepper et al. LeanVec: Search Your Vectors Faster by Making Them Fit. TMLR (2024).

[5] Jaasaari et al. LoRANN: Low-Rank Matrix Factorization for Approximate Nearest Neighbor Search. NeurIPS (2024).

[6] Chen et al. RoarGraph: A Projected Bipartite Graph for Efficient Cross-Modal Approximate Nearest Neighbor Search. VLDB (2024).

[7] Hyvonen et al. A Multilabel Classification Framework for Approximate Nearest Neighbor Search. JMLR (2024).

[8] Wei et al. Robust Tree-based Learned Vector Index with Query-Aware Repartitioning. KDD (2025).

---

> ### Author Rebuttal · Authors · 2026-03-30
>
> We thank you for the detailed review. Some of your criticism was pointed towards the fact that our experiments lack alternative indexing methods/encoders.
>
> ⭐ Please find **[HERE](http://tiny.cc/qp41101)** a link to new plots answering these requests⭐
>
> Here are a few more clarifications (edited for space).
>
> > **While SupportNet is conceptually neat…it seems inferior…to KeyNet.**
>
> As noted by `R. F9G1`, we see them as dual formulations:
> - For **routing** problems, as $c\gg 1$, `SupportNet` is the only tractable approach, as `KeyNet` uses too many parameters to handle output dimension $c\cdot d$, see **L.383, 2nd col.**
> - For **direct key prediction**, `KeyNet` is preferable, **L.413**.
>
> > **…using some approximation is of course not new or surprising. It is not convincing that…a neural network…for predicting the optimal key would be preferable to…directly predicting the clusters to search [1] or speeding up the within-cluster search…[4, 5]**
>
> Thanks for the references, we have added them. A few clarifications:
>
> - For routing, [1] proposes a linear *classifier* (very different from our regressors) and does not discuss FLOPs. Fig.2 (S.4.4) shows our models significantly outperform centroid routing at iso-FLOP levels (Pareto fronts).
>
> - For within-cluster (DB) search (S.4.5, $c=1$), it was our intention to use IVF as a backbone indexer, fed with either $\mathbf{x}$ **or** $\hat{\mathbf{y}}(\mathbf{x})$ **L.426**, and we expected other backbones to yield similar results. **We understand your skepticism.** ⭐ Please look at **§3** for eval plots using {SCANN, SOAR, LeanVEC}, beyond FLOPs (latency (ms) and #probes).
>
> > **note that the latter ones also adapt to the query distribution**
>
> Spot on — **L.37, 2nd column** was very misleading; we now write:
>
> *While effective, these approaches make no or limited use of the query distribution (e.g. by optimizing quantization lookups or criteria involving linear transforms).*
>
> We erased the last sentence in **L.94-97 (col right)** and expanded discussion.
>
> > **it is not clear how they would extend to $k$-MIPS…**
>
> - In *theory*: $f_k(x):=\max_{i_1, \dots,i_k; i_j\ne i_\ell} \sum_{j=1}^k \langle x, y_{i_j}\rangle$ is convex. The top-k key can be retrieved as $\nabla f_{k}(x)-\nabla f_{k-1}(x)$…we left it for future work.
> - In *practice*: search top-$k$ matches around $\hat{\mathbf{y}}(\mathbf{x})$.
>
> >  **…set the number of clusters as c = 10 which is much lower…value should be $O(\sqrt{n})$ (e.g. [2])**
>
> This remark conflates our two experiments:
>
> - In S4.4, $c=10$ shows that neural routing beats centroid-based routing. Both use $c=10$ + exhaustive within-cluster search, so the comparison is fair, regardless of whether $c=10$ is optimal. ⭐ Please look at **§8** of new plots. We have ran a `XS`  `SupportNet` on `NQ` using $c=128$ clusters (trained on 1B samples). `KeyNet` would not be tractable, as its output dimension would need to be $128\times 384 = 49,152$ vs. a $P=9M$ total params count (1% of $n$=2.5M x $d$=384).
>
> - In S4.5, $c=1$ for our models; **we use the $O(\sqrt{n})$ default for the FAISS/IVF index.**
>
> > **use only one embedding model**
>
> ⭐ Please look at **§1** of new plots. We added `DistillRoberta` and `MPNET` ($d=768$). Our models train out of the box.
>
> > **The used embedding model produces normalized vectors…**
>
> This is correct, but all modern bi-encoder retrievers in NLP (DPR, BEIR baselines, sentence-transformers) output normalized embeddings.
>
> Crucially, none of our architectures nor training loops use normalization (Homogeneization is unrelated to normalization). We tried leveraging normalization (e.g. Eikonal regularizer, post-hoc normalization) but saw no gains.
>
> > **…not evident that the query and database distributions actually differ**
>
> ⭐ Please look at **§6**. providing illustrations to support that they differ, especially for larger `NQ` and `HotPotQA`. We now believe that the larger the query/key distribution gap, the better our methods perform
>
> > **How much training data is used in your experiments and long does training the networks take…which hardware?**
>
> Queries are augmented 5–100X **(L. 280)** , e.g.:
> - Quora: 0.5M queries × 100 aug. = 50M
> - HotPotQA: 5M queries × 10 aug. = 50M
> We train on 1B–10B samples **(Fig. 7) x-axis**, i.e. 20–200 epochs, on H100 nodes: Takes 6 hours (XS, 1B) to 1 week (XL, 20B).
>
> ⭐ Please look at **§4** and **§7** for corrected Fig. 7 on y-axis.
>
> > **How much effect do the different terms in the loss functions actually have**
>
> ⭐ Please look at **§2**.  Our latest view is that `SupportNet` for routing can be trained using only scores; For `KeyNet`, we stand by **L.308**
>
> > **clarify the setup of the experiment in S.4.5…whether the number of clusters was also set to 10**
>
> No, we use FAISS' default.
>
> > **do you query the actual IVF index using the predicted vector (with no reranking based on the original query) or just choose the clusters based on the predicted vector?**
>
> We query IVF using only predicted vector **L.426**.

---

> > ### Author Rebuttal · Reviewer_wt6o · 2026-04-03
> >
> > Thank you for your detailed reply. However, I still have concerns regarding the response.
> >
> > Unfortunately I am unable to open the external link for the additional experiments as you have used an URL shortener that is blocked for me. However, I will assume that all shown results are positive.
> >
> > > For routing problems, as $c \gg 1$, SupportNet is the only tractable approach, as KeyNet uses too many parameters to handle output dimension $c \dot d$, see L.383, 2nd col.
> >
> > Thank you for the clarification! For the approximate search integration in Section 4.5, it was clear from the paper that KeyNet is the preferred approach but the difference for routing in Section 4.4. was less clear as both methods seem comparable in Figure 2. The example below with $c = 128$ shows the difference well. I would suggest highlighting this already in the introduction.
> >
> > > While effective, these approaches make no or limited use of the query distribution (e.g. by optimizing quantization lookups or criteria involving linear transforms).
> >
> > This is accurate for [4] and [5], but note that e.g. the RoarGraph method takes the query distribution into account effectively.
> >
> > > In S4.4, $c = 10$ shows that neural routing beats centroid-based routing. Both use $c = 10$ + exhaustive within-cluster search, so the comparison is fair, regardless of whether is optimal.
> >
> > I didn't mean to imply that the comparison is not technically fair. The point I was making is that typically with an optimal number of clusters the number of total points you'd expect to scan (for $k = 1$, on large datasets) is less than 1% of the size of the dataset which is already the size of your smallest considered model.
> >
> > > In S4.5, $c = 1$ for our models; we use the default $O(\sqrt{n})$ for the FAISS/IVF index.
> >
> > This seems more reasonable, but it would be helpful to know the actual value. Faiss does not have default values for the number of clusters.
> >
> > > This is correct, but all modern bi-encoder retrievers in NLP (DPR, BEIR baselines, sentence-transformers) output normalized embeddings.
> >
> > While most models do indeed output normalized embeddings, this is not true always, see e.g.
> > https://www.sbert.net/docs/pretrained-models/msmarco-v5.html
> >
> > More importantly, I don't believe that your paper presents itself as being applicable only to embeddings. MIPS is also important in e.g. recommender systems (as you state in your introduction). It would be preferable to evaluate on common datasets used in the MIPS literature.

---

> > > ### Author Response · Authors · 2026-04-04
> > >
> > > Many thanks for reacting to our rebuttal.
> > >
> > > We are sorry that you encountered difficulties accessing our content. We used an **anonymity preserving** link shortener (we did not log into any account to create it, cannot track it) to avoid wasting characters in responses.
> > >
> > > Here is the original link:
> > > https://docs.google.com/document/d/e/2PACX-1vR-aMDZ38th_dUxGV5Zk631Lj2afYoVNNkMfbIwixSlLI3xf8WtNBkfRyB9uiRgcLVT2fAGkSOEShoC/pub
> > >
> > > If you do not have access to the link above, you may download this (anonymized, no metadata) pdf
> > > https://1drv.ms/b/c/e8c2d340023f939c/IQBV04qdSJthRLugTiy1bkivAYI0uYCAGn92ymp18x__1_M?e=6iFRzz
> > >
> > > **If this still does not work, please let us know through the AC, we must find a way, as many of your questions are answered in that document**
> > >
> > > > **[routing problems] Thank you for the clarification! ... I would suggest highlighting this already in the introduction.**
> > >
> > > Thanks for this suggestion! We will emphasize the duality between learning a score to do routing vs learning the top-1 key to do query transformation earlier in the paper in **L71**.
> > >
> > > Additionally, we will mention in the conclusion that they can be used complementarily: a `SupportNet` router for clusters, a `KeyNet` for each cluster, sharing no backbone to distribute training more easily.
> > >
> > > > **[References] RoarGraph method takes the query distribution into account effectively.**
> > >
> > > Thanks again for your references. Of course, our goal is to give proper credit to all existing works. As mentioned, we will gladly add these references, and rewrite our introduction and related work (p.2 first pargraph). RoarGraph is indeed a very relevant reference to add, as it discusses reasons why OOD queries might degrade the performance of current indexing approaches.
> > >
> > > What we meant by taking the "query distribution" into account is that we formulate our amortization problem as a *(convex) regression* with an *expectation* $\mathbb{E}_{x\sim\mathcal{Q}}$ where $\mathcal{Q}$ is a distribution (**L220-260**). This perspective facilitates, for instance, ideas such as using hundreds of data augmentations to "learn" better regressors.
> > >
> > > Taken together,
> > > - our use of neural networks
> > > - our amortization / loss minimization perspective
> > >
> > > truly make our contribution orthogonal (and in fact complementary!) to all references you have listed, while at the same time being completely within the scope of ICML.
> > >
> > > > **typically with an optimal number of clusters the number of total points you'd expect to scan (for $k=1$, on large datasets) is less than 1% of the size of the dataset which is already the size of your smallest considered model.**
> > >
> > > There might be a misunderstanding. For the routing experiment, the cost of comparing a query to $c$ centroids is truly negligible compared to running our smallest `XS` models, we 100% agree with you on that.
> > >
> > > The point we want to make is that this neural routing mechanism, while more costly, is significantly more accurate than centroid comparison, while returning estimates for *all* clusters in *one* evaluation. Our convex regression approach to learn support functions can help find easily which clusters match best with a query. One can easily handle the addition of new clusters, by simply training as many support functions (without changing the rest, something a classifier cannot handle)
> > >
> > > For instance (if you are not able to access our files), in the last plot of our rebuttal (with $c=128$, `NQ`),
> > > - the *top-1* cluster returned by our method is accurate 72% of the time. the *top-1* cluster returned by best matching centroid is only correct 56%. This means that almost half of the time, compute would be wasted doing exhaustive search within the wrong cluster.
> > > - searching within our *top-4* clusters is ~95% accurate, while *top-16* centroids is ~96 using 4X more FLOPS.
> > >
> > > > **This seems more reasonable, but it would be helpful to know the actual value.**
> > >
> > > We hardcoded round numbers that are close to $\sqrt{n}$. We used
> > > - `1000` for `Quora`, (0.5M)
> > > - `1250` for `Nq` (2.5M)
> > > - `2000` for `HotPotQa` (5M)
> > > - `4500` for `BioAsq` (15M)
> > >
> > > > **While most models do indeed output normalized embeddings[...]**
> > >
> > > While adding more datasets is of course always desirable, we chose to use our compute on the BEIR datasets, widely accepted by the ICML community (https://arxiv.org/abs/2104.08663, ~1700 citations)
> > >
> > > > **More importantly, I don't believe that your paper presents itself as being applicable only to embeddings...**
> > >
> > > **If this aspect is important to the reviewer**, we commit (and we have absolutely no problem doing so) to mentioning earlier, as early as in the second paragraph of the introduction, that our area of interest concerns *MIPS for embeddings*. We believe that MIPS for embeddings is, in itself, crucial on its own (e.g. RAG for LLMs). **Please look at our suggested changes to abstract (see final response to Reviewer `9vKD`)**
> > >
> > > Thanks again for your time, your valuable feedback has helped improved our submission.
> > >
> > > Authors

---

### Official Review · Reviewer_F9G1 · 2026-03-13

**Soundness:** 3
**Presentation:** 3
**Significance:** 3
**Originality:** 4
**Overall Recommendation:** 6
**Confidence:** 4

**Summary:**

This paper proposes amortized MIPS: a learning-based framework that trains neural networks to directly predict solutions to Maximum Inner Product Search (MIPS) problems, exploiting the mathematical fact that the MIPS value function is the support function of the key set, a convex, positively 1-homogeneous function whose gradient equals the optimal key. Two architectures are introduced: SupportNet, an input-convex neural network (ICNN) that models the support function (with optimal keys recovered via autodiff gradients), and KeyNet, which directly regresses the optimal key vector. Both extend to a multi-task clustered setting for use as a routing mechanism. Experiments on BEIR benchmarks (FIQA, Quora, NQ, HotpotQA) demonstrate improved routing accuracy over centroid baselines and improved recall-vs-FLOPs tradeoffs when integrated with FAISS IVF indices.

**Compliance With Llm Reviewing Policy:**

Affirmed.

**Final Justification:**

This paper proposes a very clean framework for solving the MIPS problem, with active discussion during the rebuttal phase. The authors provided a substantial empirical result to support their claims and completely clarified the questions and concerns.

**Key Questions For Authors:**

1. How does performance scale with embedding dimensionality beyond d=384? The quadratic growth of parameter count with d (via the $h^2$ term) could become prohibitive at larger scales.

2. What is the total wall-clock training time (including pre-computation) compared to building a HNSW or ScaNN index of equivalent size? Without this, the "amortization" argument is hard to evaluate concretely.

3. For the approximate search integration (Section 4.5), can you provide a comparison against at least one learned quantization baseline (e.g., ScaNN or OPQ) on the same FLOPs budget?

4. Can the authors provide an ablation isolating the contribution of the homogenization wrapper and the score consistency / gradient matching losses individually?

**Limitations:**

1. The method fundamentally assumes access to a representative, fixed query distribution at training time. Many real retrieval systems face non-stationary or long-tailed query distributions for which this assumption may fail silently.

2. The method may be prohibited by billion-scale databases. However, the reviewer appreciates the authors' efforts in making significant algorithmic contributions.

3. While the framework is theoretically motivated, there are no formal bounds on the approximation of support function or the resulting retrieval error.

**Strengths And Weaknesses:**

Overall, the reviewer believes that the authors provide significant contributions to the MIPS problem. The originality of the work is highly appreciated, and the mathematical reasoning is very clean.

Strengths:
1. Connecting MIPS to support functions from convex analysis is insightful and clean.
2. The SupportNet/KeyNet duality is well-motivated: one prioritizes mathematical grounding (convexity, homogeneity), the other prioritizes inference efficiency. The tradeoffs are analyzed honestly.
3. Grounding the KeyNet auxiliary loss in Euler's theorem for homogeneous functions is a theoretically motivated and novel regularization idea.
4. The FAISS IVF integration in Section 4.5 demonstrates that predicted keys meaningfully improve recall-vs-compute tradeoffs on a real large-scale dataset (HotpotQA, 5.2M keys).
5. Connection to optimal transport.

Weaknesses:
1. Embedding dimensions are not thoroughly experimented.
2. Some missing baselines: DiskANN, ScaNN, etc.
3. Training cost is not discussed.
4. Presentation, especially figure annotations (axis labels, legends, etc.) could be improved.

---

> ### Author Rebuttal · Authors · 2026-03-30
>
> We are sincerely grateful for your time reading our paper and for your detailed review. Your appreciation for our work is uplifting!
>
> ⭐ Please find **[HERE](http://tiny.cc/qp41101)** a link to new plots ⭐
>
> > **[W1] Embedding dimensions are not thoroughly experimented … [Q1] How does performance scale with embedding dimensionality beyond d=384?**
>
> ⭐ Please look at **§1** of new plots, in which we explore `KeyNet`'s performance using two classic $d=768$ encoders `RoBERTa` and `MPNet`. We trained our models out of the box, with *exactly* the same hyperparameters as used with the $d=384$ `all-MiniLM-L6-v2` encoder. Performance gains remain.
>
> > **The quadratic growth of parameter count with d (via the $h^2$ term) could become prohibitive at larger scales.**
>
> This is a misunderstanding. The parameter count does not grow quadratically with $d$. The parameter count $P$ (**L.174**) is fixed beforehand by the end-user as a fraction of dataset size, $P=\rho n d$, where $\rho\leq 1$.
>
> Network sizing **L.217** start from $P$, using $L$ the depth of the network (user defined), and, optionally $d_\text{out}$ when using a multi-cluster output (e.g. in S.4.4 we used $c=10$).
>
> The width of intermediate layers $h$ is computed from $n,d,\rho,L$ (Eq. 5). The square-root terms in the numerator grows linearly with $d$. The discussion on $h^2$ (hidden state $h\times h$ MLP layers) only refers to using low (shallow) or large (deep) $L$.
>
> > **[W3] Training cost is not discussed. [Q2] total wall-clock training time vs HSNW…**
>
> We agree this was missing. We mentioned in **(L. 280)** that the training set was augmented 5–100X, e.g.:
> - Quora: 500k queries × 100 aug. = 50M
> - HotPotQA: 5M queries × 10 aug. = 50M.
>
> We train typically on 1B–10B samples **(Fig. 7) x-axis**, i.e. 20–200 epochs, on 8xH100 nodes. This can take anywhere from 6 hours (XS, 1B) to 1 week (XL, 20B).
>
> ⭐ Please look at **§4** in new results: while training metrics keep on improving with longer horizon, this does not necessarily impact search cost/performance tradeoffs beyond 3B samples (typically ~1 day on a H100 node). Note also that we correct in **§7** in the y-axis values of Fig. 7 in the submission.
>
>
> > **[W2] Some missing baselines: DiskANN, ScaNN, etc. …[Q3] a comparison against at least one learned quantization baseline (e.g., ScaNN or OPQ)**
>
> ⭐ Please look at **§3**. We followed your (and other reviewers) suggestions and added `ScaNN`, `SOAR` and `LeanVEC`. We report results as a function of FLOPS, latency and num probes.
>
> **Food for thought**: While `KeyNet` still provides a nice edge in terms of latency when used with these new query-optimized indexers, for the *very best* integration, **leveraging the best of both worlds**, one might prefer running ScaNN indexing and quantization optimization using **our learned predictions** (and *not* the original queries, as we did).
>
> > **[Q4] provide an ablation isolating homogenization wrapper … the score consistency / gradient matching**
>
> ⭐ Please look at **§2** for ablations regarding losses. They work as intended: for the same model, increasing weight of score (gradient) loss yields lower score (gradient) error.
>
> For target prediction (S.4.5), we believe our loss mix is good. We had observed that adding a bit of score loss would improve score error, with no degradation in gradient error.
>
> For the routing task (S.4.4 in submission), we are shifting our view: we think that it might be better  to focus more (or exclusively, as we did in **§8**) on score loss for small `XS` networks.
>
> For Homogenizer wrapper, our experience running jobs suggests a more stable training behavior, but we were not able to collect conclusive evidence during the rebuttal period. We will work on this further. Note that because all datasets are currently normalized at inference, this only changes training dynamics but plays no role in evals.
>
> > **Many real retrieval systems face non-stationary or long-tailed query distributions for which this assumption may fail silently.**
>
> Now that we propose a ML view on MIPS, we can think of classic "learning" ways to mitigate these issues (e.g. continual learning, detection of dataset shifts, augmentations etc.).
>
> We also note classic indexing methods would also see degradation in their performance/compute curves in such scenarios.
>
> > **The method may be prohibited by billion-scale databases.**
>
> Many challenges arise when going at larger scales, notably dataloaders and precomputation of ground-truth assignment. However, we do believe our learning paradigm can scale.
>
> ⭐ Please look at **§3**-FAISS-**BioAsq** results. BioAsq has 15M keys.

---

> > ### Author Rebuttal · Reviewer_F9G1 · 2026-04-02
> >
> > The reviewer thanks the authors for their detailed answers. The additional results completely clarify previous questions and concerns. Score remains 6.

---

> > > ### Author Response · Authors · 2026-04-04
> > >
> > > Dear Reviewer,
> > >
> > > We are happy to hear that our rebuttal was useful. While we are obviously very grateful for your early support of our submission, we value your comments even more. We believe they have helped us improve substantially our submission.
> > >
> > > Many thanks again for your time and for your interest in our work.
> > >
> > > The Authors

---

### Decision · Program_Chairs · 2026-04-30

**Decision:**

Accept (regular)

**Comment:**

The paper introduces Amortized Maximum Inner Product Search (MIPS), a learning-based framework that shifts the computational burden of search from the query phase to a training phase.

Reviewers appreciated the creative use of the convexity of the MIPS value function to formulate a learning problem.

Initial reviews expressed skepticism regarding whether a neural network regressor would be more efficient than established clustering or indexing techniques (e.g., predicting clusters directly). Reviewers also requested performance data beyond FLOP counts, specifically focusing on actual latency and the number of probes. There were questions about how well the method integrates with different indexing backbones like IVF.

In the rebuttal phase, the authors successfully addressed several core concerns: The authors provided updated evaluation plots using diverse backbones such as SCANN, SOAR, and LeanVEC, moving beyond FLOPs to measure real-world latency (ms) and probing efficiency. They clarified that their regressor-based approach is fundamentally different from standard linear classifiers and provides superior results at comparable computational costs.

The authors revised misleading sections of the manuscript to better explain how the model adapts to query distributions.

While some skepticism remains regarding the overhead of training such models for smaller databases, the consensus is that the method represents a significant and mathematically grounded step forward in learned indexing.

The strong empirical results on Pareto fronts and the successful integration with various backbones justify its acceptance.